# Solubility characteristics of soil humic substances as a function of pH: mechanisms and biogeochemical perspectives

Xuemei Yang[1,2], Jie Zhang[1], Khan M.G. Mostofa[1,3*], Mohammad Mohinuzzaman[1,4], H. Henry Teng[1,3], Nicola Senesi[5], Giorgio S. Senesi[6], Jie Yuan[7], Yu Liu[1,3], Si-Liang Li[1,3], Xiaodong Li[1,3], Baoli Wang[1,3], and Cong-Qiang Liu[1,3*]

[1]School of Earth System Science, Tianjin University, 92 Weijin Road, Tianjin 300072, China.
[2]Institute of Ecology, College of Urban and Environmental Sciences, Peking University, Beijing, 100091, China
[3]Tianjin Key Laboratory of Earth Critical Zone Science and Sustainable Development in Bohai Rim, Tianjin University, Tianjin 300072, China
[4]Department of Environmental Science and Disaster Management, Noakhali Science and Technology University, Noakhali, Bangladesh.
[5]Dip.to di Scienze del Suolo, della Pianta e degli Alimenti, Università degli Studi di Bari "Aldo Moro", Via G. Amendola 165/A, 70126 BARI – Italy.
[6]CNR - Istituto per la Scienza e Tecnologia dei Plasmi (ISTP) - sede di Bari Via Amendola, 122/D - 70126 Bari, Italy.
[7]College of Resources and Environment, Xingtai University, Quanbei East Road 88, Qiaodong District, Xingtai City, Hebei Province.

*Corresponding author:* Khan M.G. Mostofa or Cong-Qiang Liu (mostofa@tju.edu.cn or liucongqiang@tju.edu.cn)

**Abbreviations**
DOC: Dissolved organic carbon
DOM: Dissolved organic matter
EEM: Excitation-emission matrix
HS: humic substances
HA: humic acids
FA: fulvic acids
PLS: protein-like substances
PLF: protein-like fluorescence
HLF: humic-like fluorescence EF: electrochemical force
IF: intramolecular force
$EF_N$: net EF
$IF_N$: net IF
LS: Labile state
CS: Complexed state
$HS_{LS}$:  LS HS
$HS_{CS}$: CS HS
$HA_{LS-pH6}$: LS HA deposited at pH 6
$HA_{CS-pH6}$: CS HA deposited at pH 6
$HA_{LS-pH1}$: LS HA deposited at pH 1
$HA_{CS-pH1}$: CS HA deposited at pH 1
$FA_{LS}$: LS FA
$PLS_{LS}$: LS PLS
$FA_{CS}$: CS FA
$PLS_{CS}$: CS PLS
SOC: Soil organic carbon

SOM: Soil organic matter

**Abstract**
Soil humic substances (HS) typically alter their electrochemical behaviors in the pH range of 1–12, which
simultaneously regulates the stability of organo-minerals by modifying the HS functionalities. This process
facilitates both biotic and abiotic transformations, which consequently leads to the export of degradative
byproducts (e.g. HS components, nutrients) from soils into surrounding aquatic environments through water
and/or rainwater discharges. However, the solubility features, environmental consequences, and mechanisms
of HS, including humic acids (HA), fulvic acids (FA), and protein-like substances (PLS), under different pHs
remain unclear. To respond to these issues, we used two soil extracts which were fractionated in the pH range
from 12 to 1. The pH-dependent presence or absence of fluorescence peaks in the individual HS components
reflected their functional group proton/electron exchange features at both low and high pH values, which
were related to their solubility or insolubility. In particular, alkaline pH ($\geq$ pH 9) yielded the anionic forms
(–O– and –COO–) of phenolic OH and carboxyl groups of $HA_{CS}$ resulted in decreased electron/proton
transfer from HS functionalities, as indicated by the decline of fluorescence peak maxima, whereas the
protonic functionalities (e.g., $-COOH$, $-OH$) of HS at lower pH resulted in the formation of highly available
and remains uncomplexed HS forms. The solubility of HA fractions increases with increasing pH, whereas
their insolubility increases with decreasing pH, which determines their initial precipitation at pH 6 and final
precipitation at pH 1, amounting approximately to 39.1-49.2% and 3.1-24.1% of the total DOM, respectively,
in the two soils. Elemental anlysis results demonstrated that the C and N contents of $HA_{LS-pH6}$ were lower
and those of O, S and H higher than those of $HA_{CS-pH6}$, suggesting the preservation of C and N without S
acquisition in $HA_{CS-pH6}$ possibly because of their complexed with minerals, which, in turn, would determine
the insolubility of the $HA_{CS-pH6}$ fraction. $FA_{CS}+PLS_{CS}$ showed relatively higher C and S contents, and lower
O% with respect to $FA_{LS}+PLS_{LS}$, impling that $FA_{CS}+PLS_{CS}$ would remain under mineral protection. Fourier
Transform Infrared (FTIR) results show significantly reduced infrared absorptions (e.g. 3300–3600 and 800-
1200 cm$^{-1}$) of $HA_{CS-pH6}$ with respect to $HA_{LS-pH6}$, suggesting the existence of strong intermolecular
interactions among HA functional groups, possibly due to insoluble forms originally complexed with
minerals. But $FA_{LS}+PLS_{LS}$ exhibited stronger bands at 3414-3429 cm$^{-1}$ and 1008-1018 cm$^{-1}$ than
$FA_{CS}+PLS_{CS}$, implying a strong interaction among functional groups possibly derived from various organo-
mineral complexes in $FA_{CS}+PLS_{CS}$. These results would indicate that HS insolubility arises via organo-metal
and organo-mineral interactions at alkaline pH, along with HApH6 insolubility via rainwater/water discharge,
whereas $HA_{pH2}$+FA+PLS appears to be soluble at acidic pH, thereby being transported in ambient waters via
rainwater/water discharge and groundwater infiltration. Therefore, the pH-dependent behaviour of soil HS
greatly contributes to a better understanding of the progressive transformation, mobility/transportation, and
immobility/accumulation of HS components under various environmental conditions, with relevant
implications for sustainable soil management practices and soil DOM dynamics.

**Key words:** Paddy and maize soils; humic acids; fulvic acids; protein-like substances; acidic-alkaline pH;
EEM-PARAFAC analysis; Fourier Transform Infrared

## 1 Introduction

Soil organic matter (SOM), especially its more chemically active components, that is humic substances (HS), are particularly important because they play a number of fundamental roles, including the control of soil fertility, climate regulation and ecosystem stability (Harden et al., 2018), plant mineral nutrition and growth (Canellas and Olivares, 2014; Schmidt et al., 2007; Trevisan et al., 2010), adsorption/desorption of trace metals and radionuclides, (Boguta et al., 2019; Bryan et al., 2012; Chou et al., 2018), and soil structural stability and porosity (Bronick and Lal, 2005; Senesi and Plaza, 2007). Loss of soil organic carbon (SOC) is due to several biotic and abiotic processes, (Crowther et al., 2016; Huang and Hall, 2017) including heterotrophic respiration (Bond-Lamberty and Thomson, 2010; Heitmann et al., 2007; Klüpfel et al., 2014; Huang and Hall, 2017) and increasing temperatures due to climate change (Davidson and Janssens, 2006). SOC loss is also affected by soil erosion caused by deforestation, tillage, and other natural degradation processes, including hillslopes, salinisation, waterlogging, and wildfires (Ellerbrock et al., 2016; Peinemann et al., 2005; Steinmuller and Chambers, 2019; De la Rosa et al., 2012; Drake et al., 2019). In general, HS are divided according to their water solubility at various pH values into humic acids (HA), which are insoluble at pH < 2; fulvic acids (FA) and protein-like substances (PLS), which are soluble under both acidic and alkaline conditions; and humin, which is insoluble at any pH (Zhang et al., 2023; Senesi and Loffredo, 1999; Mohinuzzaman et al., 2020).

Three-dimensional (3D) fluorescence excitation-emission matrix (EEM) spectroscopy (3D EEMS) is a precise, rapid and relatively simple technique for measuring filtered environmental surface waters and samples extracted from soils and sediments (Senesi, 1990a; Coble, 1990, 1996; Stedmon et al., 2003; Mostofa et al., 2013; Mohinuzzman et al., 2020). In particular, this technique allows the characterization of fluorescent components, including soil HS, autochthonous humic-like substances, PLS, detergent-like substances, and others, without the need for further pretreatment of the samples (Senesi, 1990a; Coble, 1990, 1996; Stedmon et al., 2003; Mostofa et al., 2013). Recently, 3D EEM combined with parallel factor (PARAFAC) analysis has been used to identify and characterise HA, FA, and PLS (Stedmon et al., 2003; Gao et al., 2018b; Tadini et al., 2020; Mohinuzzaman et al., 2020). In particular, two typical protein-like fluorescence (PLF) peaks (T and $T_{UV}$) and a minor component consisting of one or two fluorescence peaks (M and/or A) attributable to humic-like fluorescence (HLF) were identified in PLS (Mohinuzzaman et al., 2020; Yang et al., 2024).

The HS solubility and insolubility mechanisms are associated with two key factors. First, the soil pH, which varies widely in soils worldwide (Table S1), influences the ionisation level of HS functional groups. In particular, high pH values favour anionic forms, i.e., $-COO^-$ and $-O^-$ of carboxylic acids and phenolic/alcoholic groups and, consequently, the formation of metal-HS complexes, including insoluble ones (Brady and Weil, 2008; Kleber et al., 2007; Min et al., 2014; Dynarski et al., 2020; Kirsten et al., 2021; Zhang et al., 2023). In contrast, relatively low soil pH values favour protonic forms, such as, $-COOH$ and $-OH$ of HS functionalities, which promote proton/electron exchange processes (Klapper et al., 2002; Nurmi and Tratnyek, 2002; Cory and McKnight, 2005; Yang et al., 2016; Wang et al., 2023). Furthermore, the zeta potential (ZP) of HA is minimal in the pH range 5–7, which is most likely caused by the

dissociation of acidic functional groups that prevail at lower pH values, whereas disaggregation
predominates over dissociation at higher pH values (Jovanović et al., 2013). Second, significant variability
in the pH of rainwater (Table S2) or any inflowing water can affect both the solubility/transport/mobility
and insolubility/immobilization/accumulation of soil HS. Thus, understanding the solubility/insolubility of
SOM/HS under changing pH conditions is important for understanding the global C cycle.
Earlier studies (Hemingway et al., 2019; Lützow et al., 2006; Marschner et al., 2008; Sollins et al.,
1996; Vogel et al., 2014) have not paid much attention to these issues when assessing the solubility and
insolubility of SOM/HS. For example, pH effects were studied to assess the interaction mechanisms of Fe(II)
ions with soil HA at pH values of 5 and 7 (Boguta et al., 2019), the binding of Cu and Pb to HA and FA at
pH 4-8 (Christl et al., 2005), Cu(II) binding properties of soil FA at pH 7.0 (dos Santos et al., 2020),
coagulation mechanisms of HA in metal ion solutions at pH 4.6-7.0 (Ai et al., 2020), coagulation behaviours
of HA in $Na^+$ and $Mg^{2+}$solutions at pH 3.6, 7.1, and 10.0 (Wang et al., 2013), and the disaggregation kinetics
of peat HA at pH 3.65-5.56 (Avena and Wilkinson, 2002), but not directly in water and alkali-extracted soil
HA, FA and PLS fractions. The acidic and alkaline pH conditions in the soil liquid phase alter the electronic
configuration of the functional groups of HS components, which in turn affect their complexation capacity
(Christl et al., 2005; Zhang et al., 2023; Avena and Wilkinson, 2002). The solubility and insolubility
mechanisms of the HS components under different pH conditions remain unknown. In particular, two key
fundamental questions regarding the effects of pH on HS are still unclear, that is, how the electrochemical
behaviour of soil HS components changes in the pH range of 1–12, and how these changes affect the
solubility/insolubility features of HS components and their mobilization/immobilization during rainwater
runoff and groundwater infiltration in soil.
Although the soil pH varies from 2.80 to 9.39 (Table S1), we fractionated the extracted solution in
the pH range from 12 to 1 for several key reasons: Firstly, HS-bound as organo-minerals primarily liberate
HS components and other constituents (e.g., various metals) in the liquid phase under alkaline extraction (0.1
M NaOH ≈ pH 13.0). Thus, it is crucial to understand how these HS components change their properties from
pH 12 to 1. Secondly, it is essential to investigate how HS, in particular HA-bound DOC, due to its insoluble
nature behave within the pH range from 12 to 1.
Most importantly, the solubility of HS components and their subsequent mineralization are very
relevant factor for the availability of soil nutrients and trace elements and the activity of soil microorganisms,
while their stability in organo-minerals affects negatively these processes (Malik et al., 2018; Varghese et
al., 2024; Lange et al., 1998; Gao et al., 2025; Soti et al., 2015; Yang et al., 2024; Gilbert et al., 2007; Zhang
et al., 2023). These issues are concurrently associated with the corresponding biological
fixation/sequestration of C, N and S by soil photosynthetic microorganisms (Green et al., 2019; Varghese et
al., 2024; Heckman et al., 2001; Levicán et al., 2008; Ma et al., 2021; Kelly et al., 2021; Gao et al., 2025),
and the subsequent release of extracellular polymeric substances and/or HS components in the neoformation
of fresh organo-minerals in soil (Whalen et al., 2024; Yu et al., 2020; Paul, 2016; Kallenbach et al., 2016).
Therefore, Therefore, the solubility or insolubility of soil HS components is crucial for a better understanding
of both soil management and soil carbon dynamics.
Therefore, the main objective of this study was to ascertain the solubility characteristics of soil HS
components under different pH conditions (pH 1–12) by analysing their fluorescence properties following
extraction from two different soils using either water or an alkaline solution. Water-extractable HS are
designated labile-state (LS) HS and are mostly subject to runoff from surface water and leaching from
groundwater (Mohinuzzaman et al., 2020; Gao et al., 2018a). Alkali-extractable HS are designated as
complexed-state (CS) HS and typically occur as organo-mineral and organo-metal complexes in soils
(Kirsten et al., 2021; Lalonde et al., 2012; Hemingway et al., 2019; Kleber et al., 2021). Furthermore, to
assess the electrochemical behaviour of soil HS components and their molecular-level characteristics based
on their pH-dependent solubility, we also analysed $HA_{LS/CS}$ precipitated at pH 6 ($HA_{LS/CS-pH6}$) and pH 1
($HA_{LS/CS-pH1}$) and a mixture of FA and PLS ($FA_{LS}+PLS_{LS}$ and $FA_{CS}+PLS_{CS}$) at pH 1. Another key objective
of this work was to provide a comprehensive view of the solubility and insolubility of soil HS based on the
mechanisms involved in the electronic configurational changes of HS reactive acidic functional groups, i.e.,
either in the protonic forms (e.g., −COOH, −OH) or in the anionic forms (e.g., −COO$^-$, −O$^-$) under various
pH conditions. This will provide a better understanding of soil properties and processes for sustainable
agricultural management.

**2 Materials and methods**


**2.1 Soil samples**


Soil samples were taken from two locations in China: a maize field and rice paddy field (Fig. S1). The maize
field soil is classified as calcaric fluvisol (WRB et al., 2015) and is located near the Beijing–Tianjin highway,
approximately 20 km from the city of Tianjin. The rice paddy soil is classified as fluvi-stagnic luvisol (WRB
et al., 2015) and is located near Shanghai. The two soils were cultivated for approximately 50 and 30 years,
respectively. Importantly, the rationale for selecting paddy and maize soils is based on their distinct
characteristics, i.e., paddy soils are submerged for extended periods, while maize soils are relatively less
influenced by the presence of water. Therefore, HS components of these two soil types are expected to be
altered very differently in their organo-mineral lability and stability in the pH range from 1 to 12. This study
is expected to provide useful information on soil carbon dynamics and contribute to minimize soil carbon
loss during agricultural practices. At each site, three soil subsamples were randomly collected from the top
horizon A (0–30 cm) and mixed homogeneously to produce a spatially representative sample at the field
scale. After oven drying to constant weight at 40°C, the samples were passed through a 2-mm sieve. Table
S3 provides information on the sampling sites, vegetation cover, and major physicochemical characteristics
of the two soil types.
The soil particle size was measured using the hydrometer method with a Mastersizer 3000 (Malvern, Table
S3). The soil extracts (see below) were obtained from 0.2-mm-sieved soils after mortar grinding with a pestle.

**2.2 Protocol used to extract water and alkali soluble SOM/HS**

In the first part of the experiment, the soil liquid phase was extracted from the two soils using either water or
an alkaline solution (0.1 M NaOH), which operationally represent, respectively, the water-extractable labile
state (LS) and the water insoluble alkali-extractable complexed state (CS) of SOM/HS (Mohinuzzaman et
al., 2020). The detailed extraction procedure is shown in the flow diagram in Fig. S2. Briefly, the water
extracts were obtained using ultrapure water (18.2 $M\Omega \cdot cm$, Mill-Q, Millipore) with a soil/water ratio of 1:10.
The mixtures were vortexed for 1 min in closed 500-ml brown bottles before being shaken for 24 h in an
orbital shaker (200 revs per min) at 25°C. The mixtures were then centrifuged for 20 min at 4000 rpm
(Thermo Fisher Scientific). SORVALL ST 16) for removing suspended solids. The supernatant solutions
were then filtered through a 0.45-µm glass-fibre filter (GF/F type, Shanghai Xin Ya Purification Equipment
Co. Ltd, China) to remove any remaining particulate matter. The solutions were then frozen at –20°C.
To obtain the alkaline extracts, the suspended soil residues from water extraction were sequentially extracted
under $N_2$ with a 0.1M NaOH solution at a soil residue/alkaline solution ratio of 1:10 (Fig. S2). In this case,
the mixtures were also vortexed for 1 min, shaken at 200 rpm for 3 h at 25°C, and then centrifuged for 20
min at 4000 rpm using the same centrifuge as before to remove suspended solids. The supernatant solutions
were then filtered through a 0.45-µm membrane filter (polytetrafluoroethylene membrane, PTFE, Shanghai
Xin Ya Purification Equipment Co. Ltd, China) to remove any remaining particulate matter. Under alkaline
conditions, PTFE filters are highly effective at separating solutions from particulate matter (Mohinuzzaman
et al., 2020). The remaining solid residue was extracted with a fresh alkaline solution for 3 h, and the above
procedure was repeated. The supernatant solutions were then mixed with former solutions and frozen at
−20°C for further processing. The original pH values for the water-extracted paddy and maize samples were
8.13 and 7.92, respectively, while alkali-extracted samples were 13.02 and 12.98, respectively.

**2.3 Protocol used to isolate solid HA and FA+PLS samples by acidification of water and alkaline extracts**

The second part of the experiment involved two distinct approaches. To adjust the pH from 12 to 1, aliquots
of 45 mL of water or alkaline extracts were placed in 50-mL glass bottles, and then the pH was progressively
adjusted to certain value in the range 12 to 1 by adding 0.1 and 1 mol $L^{-1}$ NaOH or HCl solutions with a 10-
µL chromatographic sampler (minimum scale 0.2 µL). As the maximum amount of acid/base reagent added
to each sample was < 1.0 mL, the dilution effect could be ignored. A Thermo Orion water quality tester,
calibrated before each measurement, was used to determine the pH of the solutions prior to further analytical
measurements. Three replicates (n = 3) were used for each pH adjustment experiment. All experiments were
performed under laboratory ambient temperature of 25°C.
In the other approach, approximately 400 mL of water extracts or alkaline extracts were placed in individual
500-mL glass bottles, the pH was adjusted to 6 using HCl (0.1 and 1 mol L$^{-1}$) and left for 10 h at 25°C to
allow the precipitation of HA$_{LS}$ and HA$_{CS}$, respectively (Fig. S2). The precipitates, denoted as HA$_{LS}$ at pH 6
(HA$_{LS-pH6}$) and HA$_{CS}$ at pH 6 (HA$_{CS-pH6}$) were separated by centrifugation (Thermo Fisher Scientific,
SORVALL ST 16) at 3000 rpm for 5 min. The supernatants were then adjusted to pH 1 using the same
procedure described above, yielding HA$_{LS}$ (HA$_{LS-pH1}$) and HA$_{CS}$ (HA$_{CS-pH1}$). The remaining supernatants at
pH 1 were classified as FA$_{LS}$+PLS$_{LS}$ and FA$_{CS}$+PLS$_{CS}$ mixtures. The HA precipitates and FA + PLS solutions
were freeze-dried before further analysis.

## 2.4 Analytical methods

The elemental compositions of the HA isolated at pH 6 and 1 and the freeze-dried mixture of FA + PLS were
measured using an elemental analyser (Elemental Vario E.L. III, Germany). Approximately 20 mg of each
dried, ground, and homogenised sample was placed in a clean, carbon-free, pre-combusted tin boat placed
on an autosampler rack assembly and loaded onto the elemental analyser. Sulfanilamide was used as the
standard after every ten measurements. The O content was calculated by difference formula: O% = 100-C%-
H%-N%-S%.
Fluorescence (excitation-emision matrix, EEM) spectra were obtained using a fluorescence
spectrophotometre (F-7000, Hitachi, Japan), as previously described (Mohinuzzaman et al., 2020; Yang et
al., 2021). To ensure instrument performance and data quality every ten samples were measured with
ultrapure (18.2 MΩ.cm) water as a blank. The water EEM spectra were subtracted from the sample EEM
spectra. A 4-µg L$^{-1}$ quinine sulfate (QS) solution in 0.01 N H$_2$SO$_4$ was used for fluorescence normalisation.
The fluorescence intensities of each sample were calibrated using the intensity of the QS (1 µg L$^{-1}$ = 1 QS
unit, QSU) peak at Ex/Em = 350/450 nm (Mohinuzzaman et al., 2020). To avoid inner-filter effects and
fluorescence quenching, the extracted solutions were diluted prior to EEM measurements based on the
initially measured DOC concentration (Tadini et al., 2018). The fluorescence intensity of each peak was
rechecked and corrected using the absorbance-based method proposed by Kothawala et al. (Kothawala et al.,

254 2013).

The pre-processed EEM data were then subjected to PARAFAC analysis using the N-way Toolbox for
MATLAB, (Andersson and Bro, 2000) as previously described (Stedmon et al., 2003). First, the Rayleigh
and Raman peaks and the ultrapure water blank spectrum were subtracted from each experimental EEM
spectrum using a homemade Excel program (Mohinuzzaman et al., 2020). To avoid mixing the fluorescent
components of different soil samples, which could produce artefacts (Mostofa et al., 2019), PARAFAC
analysis was performed individually for each selective samples. Finally, nonnegative constraints were applied
to the PARAFAC model. The detailed procedure used for PARAFAC analysis of the EEM spectra has been
described previously (Mohinuzzaman et al., 2020).
The Fourier Transform Infrared (FTIR) spectra were recorded on 2 mg aliquots of each dehydrated HA
isolated at pH 6 and 1, as well as each freeze-dried mixture of FA + PLS, which were mixed with 200 mg of
dried KBr, and pelletised by pressing under reduced pressure. The spectra were measured over the range of
$4000–400 \text{ cm}^{-1}$ by averaging 30 scans at a $4\text{-cm}^{-1}$ resolution using an IR Affinity-1S spectrometer (Shimadzu,
Japan) that included a high-energy ceramic light source, a temperature-controlled, high-sensitivity DLATGS
detector, and a high-throughput optical element, which allowed the optimisation of the electrical and optical
systems to achieve the highest signal-to-noise (SN) ratio.
**3 Results and discussion**
**3.1 Fluorescence spectra**
The fluorescence peaks of HA, FA, and PLS in the EEM spectra of the water and alkaline extracts of each
sample (original and adjusted pH) were identified individually by applying the PARAFAC model (Figs. 1,
2; Table 1). The fluorescence properties of the original samples were similar to those measured in an earlier
study (Mohinuzzaman et al., 2020), but the EEM images and fluorescence peaks of all three components
(HA, FA, and PLS) identified in the pH-adjusted samples exhibited distinct differences. Such differences
could be attributed to the pH-influenced changes in protonation/deprotonation of the each component's
functional groups, which could either suppress or favour electron transfer processes from the functional
groups to the solution (Mostofa et al., 2013; Senesi, 1990b). Two fluorescence peaks were identified in the
HA (peaks C and A) and FA (peaks M and A) components, while four peaks were identified in the PLS
fraction: peaks M and A for HLF, and peaks T and $T_{UV}$ for PLF (Mohinuzzaman et al., 2020).
**3.1.1 Characteristics of $HA_{LS\text{-}pH6/pH1}$ and solubility of $HA_{CS}$**
The EEM-PARAFAC model detected no fluorescence in water-extracted $HA_{LS}$ at acidic pH ranging from 6
to 1 (Fig. 1). This causes HA to precipitate at pH 6 ($HA_{LS\text{-}pH6}$) and at pH 1 ($HA_{LS\text{-}pH1}$), accounting for
approximately 48.3-49.2% and 3.1-10.8% of total $DOC_{LS}$, respectively based on initial $DOC_{LS}$ concentrations
of 13.4 and 24.5 mg/L, respectively in paddy and maize soils (Fig. 3). Absence of fluorescence or
precipitation at pH < 7 suggests that $HA_{LS}$ may naturally stabilise in soil during rainfall/water runoff at pH
values $\leq$ 6 due to its water insolubility. However, at higher pH levels, the fluorescence peak maxima (C: 310-
340/432-460 and A: 250-275/432-460) and intensities varied significantly (Fig. 1 and 3; Table 1). The highest
C and A peak intensities of $HA_{LS}$ were observed at pH 7, with a gradual decrease as pH increased in both
soil $HA_{LS}$. At pH 7-8 (peaks C: 325-340/432-440 nm and A: 275/432–440 nm; Table 1), the functional groups
can donate their electrons, thus increasing their fluorescence intensity, whereas the blue shift and decreasing
intensity of the fluorescence peaks with increasing pH are caused by the deprotonation/ionization of COOH
and OH functional groups. The deprotonated functional groups would form organo-metal complexes through
donation of $\pi$–electron to $d$-orbitals of metal ions particularly Fe ions, (Zhang et al., 2023) which insolubilise
HS/SOC in soil (Kirsten et al., 2021; Six et al., 2002; Lalonde et al., 2012; Hemingway et al., 2019; Kleber
et al., 2021; Makiel et al., 2022). In contrast, the red-shift of the fluorescence peaks could be attributed to
easier electron transfer from the functional groups of HA (Mostofa et al., 2013; Senesi, 1990b).
Unlike $HA_{LS}$, the excitation/emission peaks of $HA_{CS}$ at pH 1-10 in maize soil were detected at wavelengths
(C, 345-385/460-477 and A, 275-280/460-477 nm) that were longer than those of the corresponding $HA_{LS}$ at
pH 7-10 (C, 325-345/432-477 and A, 260-280/432-477 nm) (Fig. 2, Table 1). These results would suggest
that HA at pH 2 may be the insoluble form of HS bound to various minerals/metals, (Kirsten et al., 2021;
Curtin et al., 2011; Lalonde et al., 2012; Hemingway et al., 2019) whereas the longer wavelength peaks (C
and A) of alkali-extractable $HA_{CS}$ functionality remains mineral protection (Mohinuzzaman et al., 2020). The
two peak maxima at longer wavelengths (350/486 nm and 275/486 nm at pH 7-8 in $HA_{CS}$ from paddy soil
and at 380/477 nm and 275/477 nm at pH 3-4 in $HA_{CS}$ from maize soil might be ascribed to electron transfer
from thiol- and/or N-containing functional groups and/or highly aromatic ring structures in alkaline-extracted
HA (Fulda et al., 2013; Haitzer et al., 2002, 2003; Szulczewski et al., 2001), as well as to binding sites
reacting with metal ions (Wu et al., 2004a, b). These groups are significantly affected by environmental
factors and soil conditions (Jiang et al., 2015; Vidali et al., 2010). In particular, an increase in acidity might
shift peak C of soil $HA_{CS}$ from a shorter to a longer excitation wavelength but does not affect peak A detected
at pH 3-4 (C, 360-380/466-477 nm and A, 270-275/466-477 nm) and pH 5-6 (C, 340-345/469-477 nm and
A, 270-275/469-477 nm) (Fig. 2; Table 1). These results would imply that increasing the acidity promotes
electron transfer from the peak C-type functional groups of $HA_{CS}$.
The shorter emission maxima of peaks C and A in $HA_{CS}$ at pH 11-12, i.e., 458 and 458 nm, and 426 and
426,460 nm, respectively, for paddy and maize soils (Table 1), would suggest that electrons released from
$HA_{CS}$ functional groups were primarily suppressed by extreme alkalinity conditions, as they would require
higher energy, as confirmed by the appearance of peaks of decreased intensity at shorter wavelengths (Fig.
3). Thus, HA showed a higher electron transfer capacity in paddy soils than in maize soils (Xi et al., 2018).
Furthermore, the significant decrease in the two peak intensities in $HA_{CS}$ at pH 1–6 would be primarily due
to precipitation at pH 6 and pH 1, which amounted approximately to 39.1-46.4% and 3.1-24.1%, respectively,
of the initial $DOC_{CS}$ concentrations of 35.2 and 79.4 mg/L, respectively, in paddy and maize soils (Fig. 3).
The higher peak intensities at pH 1-2 than at pH 3-4 (Fig. 4) would suggest that some functional groups were
labile at this pH, thus favouring electron transfer from $HA_{CS}$ in both soils; this was also confirmed by the
longer wavelength of the excitation/emission peak C at pH 1-2.
**3.1.2 Behavior of $FA_{LS}$ and $FA_{CS}$ as a function of pH**
Both $FA_{LS}$ and $FA_{CS}$ showed higher intensities of the two peak maxima at pH 3-4, (i.e., 325-335/439-460
and 270/439–460 nm, respectively) and at pH 1-4, (i.e., 315-340/449-460 and 260-270/449–460 nm,
respectively) than at alkaline pH, with the former showing a blue shift with respect to the latter for both soils
(Figs. 1, 2; Table 1). The pH-dependent differences arising in $FA_{LS}$ might be due to existing environmental
factors (e.g. moisture, temperature/climatic warming, redox properties, mineral matrix, agricultural practices,
vegetation, and microbial activities), whereas those in $FA_{CS}$ might remain under mineral protection because
of their occurrence in organo-mineral complexes (Kirsten et al., 2021; Mohinuzzaman et al., 2020; Lehmann
and Kleber, 2015; Gao et al., 2018a). Moreover, peak M disappeared at pH 7–8 and a minor peak appeared
in the original $FA_{LS}$, suggesting degradation of the functional groups in $FA_{LS}$ at pH 7–8. The longer-
wavelength peak maxima measured at extremely acidic pH 1–4 would indicate easier electron/proton transfer
from the protonated phenolic groups in both $FA_{LS}$ and $FA_{CS}$ (Klapper et al., 2002; Nurmi and Tratnyek, 2002;
Cory and McKnight, 2005; Yang et al., 2016; Wang et al., 2023). In contrast, increasing the pH would imply
the ionisation of phenolic groups, which would necessitate more energy for the electron transfer process,
resulting in peak maxima at shorter wavelengths under alkaline conditions.
The wavelength differences detected for peak maxima were accompanied by differences in their intensity,
which was the highest for peak M of $FA_{LS}$ at pH 6 and increased by approximately 362% and 20.0%,
respectively, in paddy and maize soil $FA_{LS}$, compared with the original $FA_{LS}$. This indicates that protonated
functional groups can transfer electron more easily than deprotonated functional groups. In contrast to $FA_{LS}$,
the highest peak M intensity of $FA_{CS}$ from paddy soil occurred at pH 12 and gradually decreased to pH 5,
whereas $FA_{CS}$ from maize soil showed the highest intensity at pH 3 and decreased up to pH 9 (Fig. 4),
suggesting a difference in peak M functional groups between the two soils. These features may be ascribed
to the different environmental conditions in the two soils, that is, long-term submersion in paddy soil and a
drier state in maize soil (Mohinuzzaman et al., 2020).
Peak A intensity followed a similar trend for both soils, peaking at pH 10 and 8, then gradually decreasing
to pH 5–6 by 57% for the paddy soil and pH 3–4 by 41% for the maize soils (Fig. 4). These results suggest
that peak A functional groups in the $FA_{CS}$ of the two soils behave similarly. The highest peak intensity of
$FA_{CS}$ in the two soils was detected at pH 3; but these peaks were absent in $FA_{LS}$ (Figs. 1 and 2). These results
suggest the presence of new functional groups in $FA_{CS}$ that are absent in water-soluble $FA_{LS}$. The decreasing
intensity of peak A toward either extremely acidic (pH 1-2) or alkaline (pH 11-12) conditions suggests an
increased suppression of electron release at either very high or very low pH conditions. Notably, both $FA_{LS}$
and $FA_{CS}$ exhibited the highest solubility under acidic conditions, such as pH 3 and pH 6, respectively.
**3.1.3 Behavior of PLS as a function of pH**
Peak M of HLF in the $PLS_{LS}$ from maize soil was most prominent at acidic pH, with very low intensity at pH
7–8, and disappearing entirely at pH 9–12 (Figs. 1, 4). These results might be ascribed to the easy electron
transfer from the corresponding functional groups under acidic conditions and to the suppression of electron
release under alkaline conditions. However, this peak was completely absent in the $PLS_{LS}$ from the paddy
soil at any pH condition, possibly due to the long-term favoured hydrolysis occurring under submerged
conditions, which does not occur in the drier maize soil where this fraction is not degraded (Mohinuzzaman
et al., 2020).
The $PLS_{LS}$ samples from both soils exhibited two PLF peaks, that is, T and $T_{UV}$, at pH 7-8, with peak T
(245/303 nm) that completely disappeared at acidic pH 1-6, but was dominant at pH 9-12. This may imply a
marked influence of the pH on the ionisation of the functional groups. In contrast, the $PLS_{CS}$ from the paddy
soil showed PLF peaks (T and $T_{UV}$) in the pH range of 3 to 10, whereas in the $PLS_{CS}$ from maize soil, they
were predominant at acidic pH 1-6, appeared as minor peaks at alkaline pH 7-10 and disappeared at pH 11-
12 (Fig. 2, Table 1). Notably, $PLS_{CS}$, like $FA_{LS/CS}$, might undergo rapid electron/proton exchange reactions
that result in the appearance of predominant peak maxima under acidic conditions, whereas the disappearance
of PLF peaks at pH 11-12 might arise, similar to $FA_{LS/CS}$, from the anionic forms of PLS, which might be
involved in stable organo-mineral complexes. In this case, the submerged conditions existing in the paddy
soil are primarily responsible for the predominant occurrence of PLF peaks in the $PLS_{CS}$, whereas the drier
conditions of maize soil (high temperature and low precipitation) cause extensive degradation of the PLF
components, with the predominant presence of the HLF components. However, the significant increase in
the peak intensities of both HLF and PLF in $PLS_{CS}$ at pH 6 implies that the responsible functional groups
would remain in a protonated state (Fig. 4), which suggests a marked pH effect on the functional groups of
$PLS_{CS}$. Similar pH-influenced changes in the peak $T_{UV}$ intensities have been reported for extracellular
polymeric substances (Zhang et al., 2010).
Finally, the predominant presence of PLF and HLF components in $PLS_{CS}$ compared to $PLS_{LS}$ suggests their
origin from newly formed insoluble complexes with minerals/metals (Ciceri and Allanore, 2015; Curtin et
al., 2011; Mohinuzzaman et al., 2020; Song et al., 2016). Furthermore, the presence of a PLF peak at 240-
245/303-305 nm at pH 9-12 in $PLS_{LS}$, which was not detected in $PLS_{CS}$, supports its origin in PLS degradation
under environmental conditions. The dominant presence of the HLF peaks in both $PLS_{LS}$ and $PLS_{CS}$ may
facilitate electron transfer from the corresponding functional groups, which is a key factor in their solubility
under acidic conditions.
**3.2 Soil properties and elemental composition of HS**
The soil total carbon (STC) and soil organic carbon (SOC) in the paddy soil (14.22 and 10.82 mg/g,
respectively) were higher than in the maize soil (13.13 and 8.76 mg/g, respectively), whereas the soil total
nitrogen (STN) in maize soil (0.78 mg/g) was higher than that of paddy soil (Table 1 in Mohinuzzaman et
al., 2020). The clay and silt contents were significantly higher in the maize soil (8.6% and 57.6%,
respectively) than in the paddy soil (2.5% and 38.3%, respectively), whereas the sand content in the paddy
soil (36.0%) was higher than that in the maize soil.
The C and N contents of $HA_{LS-pH6}$ from both soils were lower than those of O, S, and H, and all atomic ratios
were higher than those of $HA_{CS-pH6}$ (Table 2). These results would suggest the preservation of C and N without
S acquisition in $HA_{CS-pH6}$ possibly because of their complex state with minerals (Hemingway et al., 2019;
Marschner et al., 2008; Vogel et al., 2014), which, in turn, determines the insolubility of the $HA_{CS-pH6}$ fraction.
In contrast, the lower levels of C and N and the high content of S that characterise $HA_{LS-pH6}$ would suggest
the degradation of the N-containing functional groups (Mohinuzzaman et al., 2020; Li and Vaughan, 2018;
Senesi and Loffredo, 1999) and the acquisition of S-containing compounds, possibly from soil fungi (Masaki
et al., 2016; Saito et al., 2002; Whelan and Rhew, 2015), which, in turn, would determine the solubility of
the $HA_{LS-pH6}$ fraction.
Due to the lack of sample $HA_{LS-pH1}$ from maize soil, no comparison was possible with the corresponding
$HA_{CS-pH1}$. However, $HA_{CS-pH1}$ from paddy soil showed extremely low C%, N%, and atomic ratios and very
high O%, H%, and S% compared to the corresponding $HA_{LS-pH1}$, indicating its insolubility at pH 1, that is,
this HA fraction would remain under mineral protection in soil (Hemingway et al., 2019; Marschner et al.,
2008; Vogel et al., 2014). It is possible that the decrease in C and increase in O in the $HA_{CS-pH1}$ fraction in
paddy soil were affected by high water availability and microbial respiration (Fang et al., 2005; Huang and
Hall, 2017; Yu et al., 2020; Chen et al., 2020).
The main features of all the FA+PLS samples were their very low C, N, C/S, C/H, and C/O ratios and very
high O%, H%, and S% with respect to the corresponding HA fractions discussed above (Table 2). In
particular, $FA_{CS}+PLS_{CS}$ showed relatively higher C and S contents and C/H and C/O ratios, and lower O%
with respect to $FA_{LS}+PLS_{LS}$, which would suggest that, similar to $HA_{CS}$ samples, $FA_{CS}+PLS_{CS}$ would remain
under mineral protection in the soil. The higher S content of $FA_{LS}+PLS_{LS}$ from paddy soil than that of maize
soil might be ascribed to the uptake and conversion of carbonyl sulfide (COS), possibly operated by soil
fungi or microorganisms in the paddy soil (Li et al., 2010; Masaki et al., 2016; Saito et al., 2002; Whelan and
Rhew, 2015), whereas S would be rapidly degraded by biotic and abiotic processes in the drier maize soil
(Liu et al., 2007; Masaki et al., 2016; Whelan and Rhew, 2015). Similarly, the relatively lower C% in
$FA_{LS}+PLS_{LS}$ and $FA_{CS}+PLS_{CS}$ from paddy soil compared with maize soil might be ascribed to extended
oxidative degradation and/or hydrolysis processes occurring in paddy soil, which lead to extended
mineralisation processes (Fang et al., 2005; Huang and Hall, 2017; Yu et al., 2020; Chen et al., 2020). Finally,
the high O% in the FA + PLS samples might have contributed to the presence of O-rich PLS extracted
together with FA.
**3.3 Fourier Transform Infrared (FTIR) spectra**
The FTIR spectra of all tested samples (Fig. 5; Table 3) were typical of soil HS (Senesi and Loffredo, 1999),
but they exhibited a number of different characteristics. First, $HA_{CS-pH6}$ had significantly lower IR absorptions
than $HA_{LS-pH6}$ in both soils, particularly in the range 3300–3600 and 800–1200 cm$^{-1}$. This suggests strong
intermolecular interactions among HA functional groups, possibly due to insoluble forms complexed with
minerals/metals (Gabor et al., 2015; Mostofa et al., 2018). This has an impact on the overall bonding system
in the conjugated macromolecular HA structure. Furthermore, these insoluble forms require relatively high
energy for electron transfer, resulting in a decrease in the relative intensity of all bands in $HA_{CS-pH6}$ compared
to $HA_{LS-pH6}$. Second, the band at 3421–3429 cm$^{-1}$ is stronger for $HA_{LS-pH1}$ than for $HA_{LS-pH6}$, indicating the
presence of more free NH or OH functional groups (Demyan et al., 2012; Kunlanit et al., 2014; Senesi et al.,
2003). Third, the weak band at 1015–1030 cm$^{-1}$ (possibly attributed to S=O and C–O–S stretching of S-
containing functional groups) in $HA_{LS-pH1}$ of the paddy soil and its absence in maize soil (Demyan et al.,
2012; Singh et al., 2011; Shammi et al. 2017; Senesi et al., 2003), might be due the degradative nature of
$HA_{LS-pH1}$ compared to $HA_{LS-pH6}$. $HA_{LS-pH1}$ degradation is primarily caused by the degradation of its functional
groups in the presence of existing environmental factors (Xie et al., 2004; Mohinuzzaman et al., 2020;
Lehmann and Kleber, 2015). Fourth, the samples $FA_{LS}+PLS_{LS}$ generally exhibited stronger bands at 3414-
3429 cm$^{-1}$ and 1008-1018 cm$^{-1}$ than $FA_{CS}+PLS_{CS}$ (Gabor et al., 2015; Kunlanit et al., 2014; Mostofa et al.,
2018), which suggested a strong interaction among functional groups possibly generated from various
silicates/mineral complexes in $FA_{CS}+PLS_{CS}$, whereas a weak interaction would have yielded free functional
groups in LS samples featuring strong bands by loosely bound electrons in functional groups. Fifth, the
presence of two relatively intense bands at 3711–3745 and 3838–3873 cm$^{-1}$ in all HA samples could be
attributed to aromatic C-H stretching in individual aromatic ring structures, while aromatic C-H in conjugated
systems absorb at 3080–3030 cm$^{-1}$ (Kunlanit et al., 2014; Demyan et al., 2012; Senesi et al., 2003).
**3.4 Mechanisms determining the insolubility/solubility of HA and FA+PLS**
Two molecular parameters, the electrochemical force (EF), also known as the intermolecular force, and the
intramolecular force (IF), are thought to control the mechanisms underlying the solubility/insolubility of HA
and FA + PLS (Fig. 6). In particular, EF includes intermolecular van der Waals forces, London forces, dipole-
dipole and ion-dipole interactions, and hydrogen bonds between molecules, whereas IF refers to the
intramolecular forces between bonded atoms in a molecule (Aeschbacher et al., 2010). In particular, the
decrease in the net EF ($EF_N$) could be attributed to the protonation of the functional groups in HA, which
decreases their electron-donating capacity in aqueous solutions (Ai et al., 2020; Chassapis et al., 2010; Ritchie
and Michael Perdue, 2003). In contrast, an increase in net IF ($IF_N$) can be attributed to increase intramolecular
interactions between various functional groups via hydrogen bonding in HA (Ai et al., 2020; Benes, 2009;
Boguta et al., 2019; Noy et al., 1997; Vezenov et al., 2005, 1997). Strong competition exists between $EF_N$
and $IF_N$; when $IF_N > EF_N$ under acidic conditions, all functional groups associate, resulting in HA
precipitation from the solution.

The solubility of $FA_{LS/CS}$ + $PLS_{LS/CS}$ at all acidic pH values was related to their higher total acidity, which
resulted from a higher number of elemental oxygen atoms (Table 2) which belong to oxygenated functional
groups and have a relatively lower molecular size than HA (Leenheer et al., 1995; Robarge, 2018). These
features would cause a relatively low $IF_N$ value and a relatively high $EF_N$ value owing to the formation of
external H-bonding with the solution components. This interpretation was supported by the presence of two
peaks for each FA and an HLF peak in the PLS at pH 1–4 (Figs. 2 and 4; Table 1). These results would
confirm the easier electron transfer from the functional groups to the solution at acidic pH, resulting in $EF_N >$
$IF_N$ implying their dissolution at extremely acidic pH (Fig. 6).
**3.5 Solubility/insolubility characteristics of soil HS and their environmental consequences**
The solubility/insolubility of the HS components was influenced by each specific pH unit, with the
involvement of various functional groups (Fig. 6) (Avena and Wilkinson, 2002; Boguta et al., 2019, 2016;
Garcia-Mina, 2006; Hernández et al., 2006) which might occur through various processes such as
complexation, ion exchange, adsorption, aggregation/coagulation, and flocculation (Avena and Wilkinson,
2002; Lippold et al., 2007; Wang et al., 2013; Jovanovic et al, 2013). In particular, (a) $HA_{CS-pH6}$/$HA_{LS-pH6}$ and
$HA_{CS-pH1}$/$HA_{LS-pH1}$ would remain in suspension under acidic conditions, whereas IF interactions preferentially
increase with increasing acidity owing to the enhanced occurrence of protonic forms of their functional
groups; and (b) the disappearance of fluorescence peaks (C, M, A, T, or $T_{UV}$) of specific functional groups
of individual HS components under any pH condition in solution would cause their interactions either with
other functional groups or coagulation/precipitation with metals or minerals (Chen et al., 2014; Helms et al.,
2013; Zhang et al., 2023; Hemingway et al., 2019; Lützow et al., 2006; Marschner et al., 2008; Sollins et al.,
1996; Vogel et al., 2014). Furthermore, each individual pH unit may sterically affect the HS functional groups
(Boguta et al., 2019; Senesi, 1990a, 1990b), which would result in either the appearance or disappearance of
a fluorescence peak and/or a change in the fluorescence intensity of specific peaks (Figs. 2, 3, 4, Table 1).
These effects may be associated with an increase or decrease in the electron donation capacity of the
fluorescent functional groups in HS (Cory and McKnight, 2005; Senesi, 1990a; Klapper et al., 2002;
Karadirek et al., 2016; Wang et al., 2023), thus determining their solubility/insolubility.
An overall conceptual model of the possible processes and mechanisms is outlined in Fig. 6 and summarised
below.
(1) The deprotonated state of the functional groups (e.g. $-COO^-$) in $HA_{LS}$ constantly donates electrons to
various soil components, thus activating a series of biogeochemical processes. Rainwater (usually at pH $\leq$ 6)
or water discharge/runoff cannot dissolve $HA_{LS}$ and, partly, $HA_{CS}$. Particularly, $HA_{LS/CS-pH6}$ that would be
insoluble/not mobile in soil during rainwater events and water runoff at pH $\leq$6, suggesting natural protection
during transport along the soil profile and in ambient surface waters. In contrast, $HA_{LS/CS-pH1}$ is mobile and
transported to ambient surface waters via rainwater, leaching, and groundwater infiltration (Ronchi et al.,
2013; Stolpe et al., 2013; Mostofa et al., 2019).
(2) Under acidic conditions, down to pH 1, the functional groups of $HA_{CS/LS-pH1}$ remained protonated, thus
reducing electron transfer capacity. This feature of $HA_{CS/LS-pH1}$ might explain some recent results, e.g. the
decline of metal binding capacity of HS at low pH (Christl et al., 2005), the low effect of HA on plant growth
(Asli and Neumann, 2010; Mora et al., 2012), the decline of HA capacity in binding organic pollutants (Jones
and Tiller, 1999; Tremblay et al., 2005), and the decrease in carbon mineralisation at low pH with a fivefold
decrease in bacterial growth and a fivefold increase in fungal growth (Rousk et al., 2009).
(3) Higher pH increases deprotonation of functional groups (e.g. $-COO^-$) of $HA_{LS/CS}$ allowing for easier
electron transfer to soil components like minerals and fungi (Chen et al., 2020; Yu et al., 2020), increasing
the solubility of metal ions (Firestone et al., 1983; Flis et al., 1993), e.g. from metal sulfides (Chou et al.,
2018), soil respiration and carbon mineralization (Pietikäinen et al., 2005; Rousk et al., 2009), and
degradation of $-COOH/-OH$ upon exposure to UV-Vis light (Spence and Kelleher, 2016; Ward et al., 2013;
Xie et al., 2004).
(4) The predominant presence of two $FA_{LS/CS}$ peaks at pH 1-2, which were absent at neutral or alkaline pH
(Figs. 2 and 4), suggests the solubility of these HS components under acidic conditions. In turn, this condition
affects the capacity for complexation/decomplexation and/or sorption/desorption of metal ions and organic
pollutants, thus modifying their mobility/transport by rainwater/water discharge/runoff and groundwater
leaching (Tadini et al., 2020; Mostofa et al., 2019) and their distribution, toxicity, and bioavailability in soil
(Anastasiou et al., 2014; dos Santos et al., 2020; Tadini et al., 2020; Zhu and Ryan, 2016). In particular, the
$FA_{LS/CS}$ fractions in acidic conditions easily leached down the soil profile via rainwater discharge, as occurs
in the podsolization process (Lundström et al., 2000).
(5) The predominance of HLF in $PLS_{LS}$ and PLF in $PLS_{CS}$ at acidic pH (Figs. 2 and 4) may be primarily
responsible for their high solubility under acidic conditions, which implies high mobility and easy transport
in ambient water environments and groundwater leaching (Gao et al., 2018a; Mohinuzzaman et al., 2020).
(6) The knowledge of the molecular-level solubility of the three HS components is essential for a better
understanding and management of agricultural practices, as affected by their individual solubility, tendency
to precipitate, and variable capability to form organo-minerals, which can occur more or less rapidly/slowly
(Underwood et al., 2024; Zhang et al., 2023). For instance, the HA fractions of acidic soils may either partially
precipitate or remain in suspension due to an increase in $IF_N$ from increasing intramolecular interactions
among various functional groups via hydrogen bonding, influenced by acidic conditions as discussed earlier.
As a result, precipitated HA fractions would enhance C stability, while suspended HA fractions are highly
prone to leaching by rainwater runoff. Furthermore, the high solubility of FA and PLS under acidic conditions
would result from prolonged water saturation occurring in paddy fields, which will lead to soil C loss by their
transport due to rainwater runoff. Simultaneously, these conditions of HS are expected to contribute to
increase the salinity levels in such type of soils (Varghese et al., 2024; Ma et al., 2021). Therefore, high-
water-demand crops such as rice may not be suitable for maintaining C stability in acidic soils. In contrast,
low-water-demanding crops like maize and wheat would be more effective in minimizing C loss from acidic
soils. On the other hand, alkaline soils can support the cultivation of a wide variety of crops while minimizing
C loss, along with the presence of relatively high levels of HS-bound to organo-minerals. In particular, the
pH levels of the paddy and maize soils object of this study are slightly alkaline (8.13 and 7.92, respectively),
making them reasonably suitable for diverse types of crops. Furthermore, both soils exhibit relatively high
levels of HS-bound organo-minerals, with significant increases in $DOC_{CS}$ stability (2.6 and 3.2 times higher,
respectively) compared to $DOC_{LS}$ lability.
(7) The strategies for addressing pH-affected soil carbon loss primarily involve the significant loss of
dissolved HS, particularly fulvic acids and protein-like fractions, from acidic soils due to water and rainwater
runoff. Therefore, it is crucial to implement an efficient, rapid and sustainable drainage system that operates
on a relatively short time scale, so that HS components may become less likely to dissolve in water and
rainwater. Effective and timely drainage from the soil surface can help prevent carbon loss from acidic soils.
Finally, the HS/SOM appeared to undergo progressive transformation under various environmental
conditions (Mohinuzzaman et al., 2020), yielding various forms of HS components (Figs. 2 and 4).
Furthermore, pH appears to control the chemical nature and electronic configuration of HA/FA/PLS
functional groups, influencing their solubility/insolubility and consequently their
mobilization/immobilization and transport/accumulation, thereby markedly affecting all biogeochemical
functions and processes in the soil. The features and extension of such processes would depend mostly on
the existing environmental conditions and factors, such as pH, soil type, organisms (e.g. bacteria, fungi, and
vegetation), temperature variations due to climate change, and precipitation frequency and intensity
(Mohinuzzaman et al., 2020; Pietikäinen et al., 2005; Rousk et al., 2009).
**4. Conclusions**
The presence, absence, or variable relative intensity of the fluorescence peaks of HS components under
different pH conditions and their relationship with electron release from their functional groups appeared to
be an excellent indicator of the HS component status. In particular, an alkaline or elevated pH level would
result in anionic forms ($-O^-$ and $-COO^-$) of phenolic OH and carboxyl groups of HA, FA and PLS, which
ultimately contributes to the insolubilisation and stability of HS through the formation of organo-mineral
complexes in soils. In contrast, at acidic pH, the electron and proton transfer processes would be facilitated
by the availability of uncomplexed metal ions, with subsequent insolubility of $HA_{LS+CS-pH6}$ which would
remain insoluble in soils during rainwater events or water runoff at pH 6, whereas $HA_{LS+CS-pH1}$ would remain
soluble and thus mobile and would be transported in ambient surface waters via rainwater, leaching, and
groundwater infiltration (Ronchi et al., 2013; Stolpe et al., 2013; Mostofa et al., 2019).
The highly soluble FA and PLS at acidic conditions would facilitate to an easy transport to ambient surface
waters via rainwater and groundwater discharge (Ronchi et al., 2013; Stolpe et al., 2013; Mostofa et al.,
2019). Furthermore, the predominant presence of PLF peaks in $PLS_{CS}$ from pH 5 to 10 in paddy soil is
indicative of solubility, whereas the relatively high degradability of $PLS_{LS}$ and $PLS_{CS}$ in maize soil may be
attributed to the dry conditions (Mohinuzzaman et al., 2020).
Finally, the insolubility of individual HS components would arise when $IF_N > EF_N$, which would be related
to the formation of hydrogen bonds between the HS functional groups and the aqueous phase, whereas the
solubility of HS components would occur when $EF_N > IF_N$. Future research directions should focus on
investigating acidic soils, which are beyond the scope of this study. These soils are expected to be
significantly affected by the individual conditions of HA, FA, and PLS in acidic environments. In conclusion,
pH was confirmed to be a very important factor in determining the solubility-insolubility of HA, FA, and
PLS in soil and should be considered with the aim of preserving soil organic carbon.

**Acknowledgement of funding sources**

This work was financially supported by the Natural Science Foundation of China (grant numbers: 42293262, 41925002, U1612441 and 42230509) and by the Key Construction Program of the National "985" Project, Tianjin University, China.

**Competing interests**

The authors declare no competing financial interest.

**Data availability**
The data collected and/or analyzed during the current study are provided in the main manuscript and supplementary materials.

**Author contribution**

K.M.G.M. and C.Q.L. designed, conceived and supervised the project; X.Y. performed the main experiments, analysed all data and prepared all Figures. J.Z. partly performed the extraction of humic acids (HA) and fulvic acids (FA) + protein-like substances (PLS) samples from soil solution and analyzes the elemental analyses and FTIR, M.M. performed the soil sampling and their preprocessing, J.Y. and X.Y. conducted the EEM-PARAFAC analysis. K.M.G.M, N.S. and X.Y. wrote the manuscript; H.H.T. and G.S.S. reviewed and edited the manuscript. All authors discussed the data and revised the manuscript.

**Table 1:** Excitation/emission (Ex/Em) wavelengths (nm) of fluorescence peaks of HA, FA and PLS identified by PARAFAC analysis applied individually to EEM spectra of original water and alkaline extracts from paddy and maize soils and of their pH-adjusted solutions at pH 1-2, pH 3-4, pH 5-6, pH 7-8, pH 9-10 and pH 11-12.

| Samples | Soil | Fluorescence peak (Ex/Em, nm) | | | | | | | |
|---|---|---|---|---|---|---|---|---|---|
| | | HA | | FA | | PLS | | | |
| | | Peak C | Peak A | Peak M | Peak A | Peak M | Peak A | Peak T | Peak Tuv |
| **Water extracts** | | | | | | | | | |
| Original(pH 8.13) | paddy | 330/467 | 270/467 | 315/439 | 230/439 | 280/409 | 220/409 | 280/335 | 220/335 |
| Original(pH= 7.92) | maize | 345/477 | 280/477 | 310/440 | 245/440 | 280/404 | 220/404 | 280/339 | 220/339 |
| pH 1-2 | paddy | nd | nd | 315/419 | 235/419 | nd | nd | nd | nd |
| " | maize | nd | nd | 330/442 | 270/442 | 305/417 | 230/417 | nd- | 230/308 |
| pH 3-4 | paddy | nd | nd | 325/439 | 270/439 | nd | 220/416 | nd | 220/305 |
| " | maize | nd- | nd | 335/460 | 270/460 | 305/422 | 230/422 | nd- | 230/304 |
| pH 5-6 | paddy | nd- | nd- | 310/442 | 265/442 | nd | 220/417 | nd | 220/303 |
| " | maize | nd | nd | 325/458 | 265/458 | 305/417 | 230/417 | nd | 230/305 |
| pH 7-8 | paddy | 340/440 | 275/440 | nd | 235/431 | nd | nd- | 275/322 | 220/322 |
| " | maize | 325/432 | 275/432 | nd | 240/423 | 285/416 | 220/416 | nd | 220/305 |
| pH 9-10 | paddy | 310/440 | 250/440 | 280/415 | 220/415 | nd- | nd- | nd | 245/305 |
| " | maize | 325/442 | 260/442 | 305/411 | 230/411 | nd | nd | nd | 245/303 |
| pH 11-12 | paddy | 325/449 | 255/449 | 305/399 | 225/399 | nd | nd | nd | 240/305 |
| " | maize | 325/460 | 260/460 | 300/416 | 230/416 | nd- | nd- | nd | 245/303 |
| **NaOH extracts** | | | | | | | | | |
| Original(pH=13.02) | paddy | 335/460 | 260/460 | 320/389 | 240/389 | 275/387 | 225/387 | nd | 225/304 |
| Original(pH=12.98) | maize | 365/460 | 275/460 | 335/451 | 245/451 | 310/405 | 235/405 | nd- | 225/304 |
| pH 1-2 | paddy | nd- | nd- | 315/449 | 260/449 | 310/398 | 225/398 | nd | 225/307 |
| " | maize | -nd | nd- | 340/460 | 270/460 | 310/416 | 225/416 | nd | 225/304 |
| pH 3-4 | paddy | 360/466 | 270/466 | 325/440 | 235/440 | 310/369 | 220/369 | -nd | 220/307 |
| " | maize | 380/477 | 275/477 | 330/440 | 240/440 | 270/386 | 220/386 | nd- | 220/305 |
| pH 5-6 | paddy | 340/469 | 270/469 | 315/403 | 230/403 | nd- | nd | 270/337 | 220/337 |
| " | maize | 345/477 | 275/477 | 325/440 | 240/440 | 310/399 | 230/399 | nd- | 230/311 |
| pH 7-8 | paddy | 350/486 | 275/486 | 300/440 | 245/440 | -nd | 220/399 | 270/339 | 220/339 |
| " | maize | 360/460 | 280/460 | 325/440 | 240/440 | 275/421 | 220/421 | nd- | 220/310 |
| pH 9-10 | paddy | 330/477 | 270/477 | 315/405 | 235/405 | -nd | 220/414 | 275/334 | 220/334 |
| " | maize | 385/460 | 275/460 | 330/440 | 240/440 | 270/404 | 220/404 | nd- | 220/305 |
| pH 11-12 | paddy | 330/458 | 265/458 | 320/388 | 240/388 | 275/387 | 225/387 | nd- | 225/303 |
| " | maize | 375/426 | 275/426,460 | 335/431 | 245/431 | nd | 220/399 | nd | 220/308 |

nd: not detected

**Table 2:** Elemental composition (%, moisture and ash free) and atomic ratios of $HA_{LS-pH6}$, $HA_{LS-pH1}$, $HA_{CS-pH6}$, $HA_{CS-pH1}$, $FA_{LS}+PLS_{LS}$ at pH 1 and $FA_{CS}+PLS_{CS}$ at pH 1.

| Sample | Soil | Ash content (%) | Elemental composition (%) | | | | | C/N | C/S | C/H | C/O |
|---|---|---|---|---|---|---|---|---|---|---|---|
| | | | C | O | H | N | C/H | | | | |
| $HA_{LS-PH6}$ | Paddy | 0.01 | 56.7 | 37.1 | 2.0 | 3.9 | 0.2 | 17 | 987 | 2.4 | 2 |
| $HA_{LS-PH6}$ | Maize | 0.02 | 54.6 | 39.6 | 2.1 | 3.3 | 0.3 | 19 | 434 | 2.2 | 1.8 |
| $HA_{CS-pH6}$ | Paddy | 0.21 | 61.2 | 32.6 | 1.7 | 4.0 | 0.1 | 18 | 1256 | 3.0 | 2.5 |
| $HA_{CS-pH6}$ | Maize | 0.41 | 58.7 | 33.3 | 1.5 | 5.6 | 0.1 | 12 | 1557 | 3.2 | 2.4 |
| $HA_{LS-pH1}$ | Paddy | 0.13 | 57.0 | 36.8 | 2.0 | 3.8 | 0.1 | 18 | 1081 | 2.4 | 2.1 |
| $HA_{LS-pH1}$ | Maize | nd | nd | nd | nd | nd | nd | nd | nd | nd | nd |
| $HA_{CS-pH1}$ | Paddy | 0.08 | 33.8 | 58.1 | 4.5 | 2.9 | 0.5 | 14 | 165 | 0.6 | 0.8 |
| $HA_{CS-pH1}$ | Maize | 0.07 | 61.6 | 32.6 | 1.3 | 4.2 | 0.2 | 17 | 904 | 4.1 | 2.5 |
| $FA_{LS}+PLS_{LS}$ at pH 1 | Paddy | nd | 35.2 | 56.8 | 4.8 | 2.2 | 0.8 | 19 | 124 | 0.6 | 0.8 |
| $FA_{LS}+PLS_{LS}$ at pH 1 | Maize | 0.76 | 37.3 | 53.1 | 5.0 | 2.8 | 0.3 | 15 | 350 | 0.6 | 0.9 |
| $FA_{CS}+PLS_{CS}$ at pH 1 | Paddy | 0.03 | 37.7 | 55.1 | 3.6 | 2.7 | 0.8 | 16 | 128 | 0.9 | 0.9 |
| $FA_{CS}+PLS_{CS}$ at pH 1 | Maize | 0.19 | 44.8 | 48.6 | 3.6 | 1.9 | 0.9 | 27 | 128 | 1.0 | 1.2 |

nd: not detected due to lack of sample

**Table 3:** Major FTIR absorption bands and assignments for HA$_{CS-pH6}$, HA$_{CS-pH1}$, HA$_{LS-pH6}$, HA$_{LS-pH1}$, FA$_{LS}$+PLS$_{LS}$ at pH 1 and FA$_{CS}$+PLS$_{CS}$ at pH 1 for paddy and maize soils.

| Wave number (cm$^{-1}$) | Assignment | HA$_{LS-pH6}$ | HA$_{CS-pH6}$ | HA$_{LS-pH1}$ | HA$_{CS-pH1}$ | FA+PLS$_{LS}$ | FA+PLSC$_{CS}$ |
|---|---|---|---|---|---|---|---|
| 3800-3750 3710-3680 | O-H stretching, -OH (free) | strong | weak | strong | weak | weak | strong |
| 3520-3500 | O-H stretching, -OH (association), N–H stretching (trace), hydrogen-bonded OH | strong | weak | strong | weak | strong | weak |
| 2930-2900 | Aliphatic C-H stretching | weak | weak | weak | strong | weak | weak |
| 2400-2200 | Nitrile C≡N | strong | weak | strong | strong | strong | weak |
| 1660-1630 | C=O stretching of amide groups(amide I band) C=O of quinone and/or H-bonded conjugated ketones | nd | nd | nd | nd | nd | nd- |
| 1600-1550 | Aromatic C=C stretching, COO— symmetric stretching | strong | strong | strong | strong | strong | strong |
| 1540-1510 | N-H deformation and C-n stretching (amide Il band), aromatic C-C stretching | nd | nd | nd | nd | nd | nd |
| 1420-1410 | C=N stretching of primaryamides (amide Ill band) | nd | nd | nd | nd | nd | nd |
| 1375-1275 | O–H deformation and C–O stretching of phenolic OH, COO— antisymmetric stretching | weak | weak | strong | strong | strong | weak |
| 1170-1120 | C-OH stretching of aliphatic O-H | nd | nd | nd | nd | nd | nd |
| 1020-1000 | C-O stretching of polysaccharides or polysaccharide-like substances, Si-O of silicate impurities | strong | weak | strong | strong | strong | weak |
| 880-780 | Out-of-plane bending of aromatic C-H | strong | weak | strong | strong | strong | weak |
| 500-450 | In-of-plane bending of aromatic C-H | weak | strong | strong | weak | weak | weak |

nd: not detected

**(a) Water extracts (LS): paddy soil**

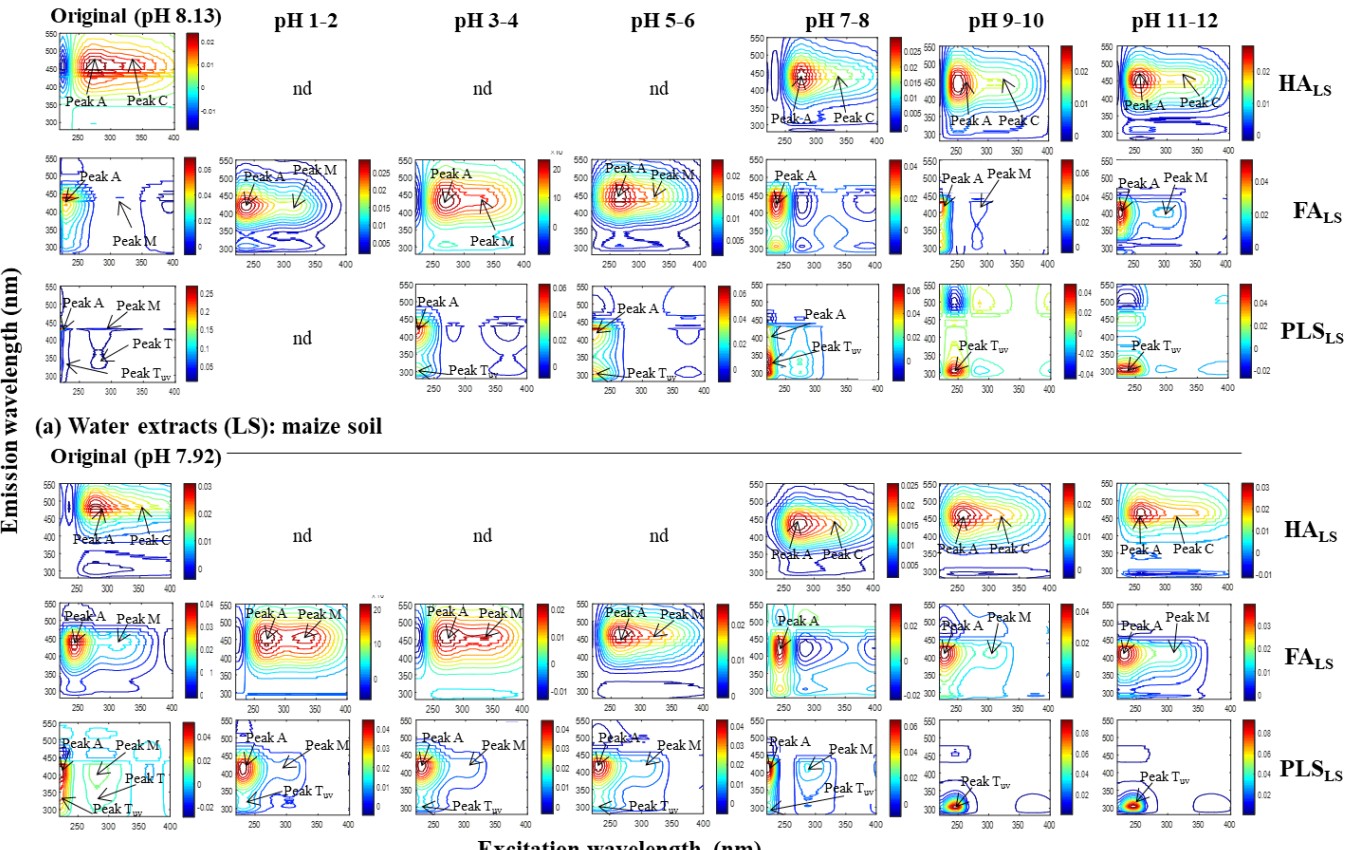

**(a) Water extracts (LS): maize soil**

**Figure 1:** Fluorescence spectra and peaks identified using EEM-PARAFAC modeling in the original solution before pH adjustment and water extracts from paddy and maize soils adjusted at various pH. Water-extractable humic acid (HA) is completely precipitated in both paddy and maize soils, as evidenced by a substantial decrease in dissolved organic carbon (DOC) concentrations of 48.3% and 49.2% at pH 6, respectively. Consequently, it results not detected (nd) by the EEM-PARAFAC model. Similarly, 'not detected' for protein-like substances (PLS) in paddy soil may be due to its highly degradative nature, as its minor peaks are detected in the original solution at pH 8.13, whereas the correspondingly functionalities disappear at highly acidic pH 1-2.

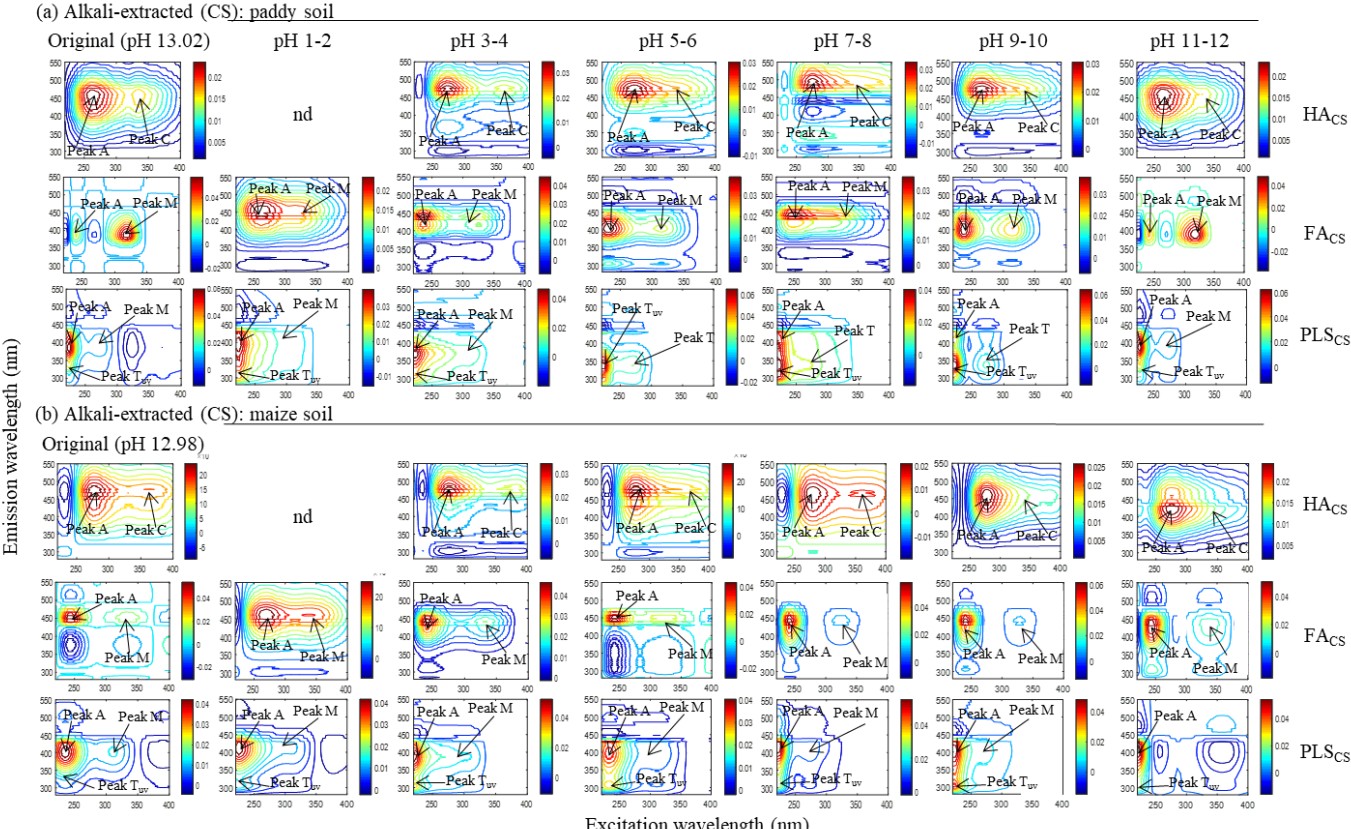

(a) Alkali-extracted (CS): paddy soil

(b) Alkali-extracted (CS): maize soil

**Figure 2**: Fluorescence spectra and peaks identified using EEM-PARAFAC modeling in the original solution before pH adjustment and alkaline extracts from paddy and maize soils adjusted at various pH. Alkali-extractable HA is completely precipitated at pH 1 in both paddy and maize soils, which is a well-known phenomenon in soil, also evidenced by a corresponding decrease of 48.1% and 53.8% of DOC concentration at pH 1, with respect to their extracted original solutions (pH ~13). Consequently, HA at pH 1-2 result 'not detected' by the EEM-PARAFAC model. Noteworthy, the detection of alkali-extractable HA at pH levels 3-6, in contrast to its absence in water-extractable HA, may result from a significantly higher release of HA from organo-minerals, which is evidenced by the DOC in alkali-extractable HA that is 2.6 to 3.2 times greater than that in water-extractable HA. In particular, water-extractable HA is highly degradative in nature and shows relatively low concentrations, which correspondingly result in its precipitation at pH 6.

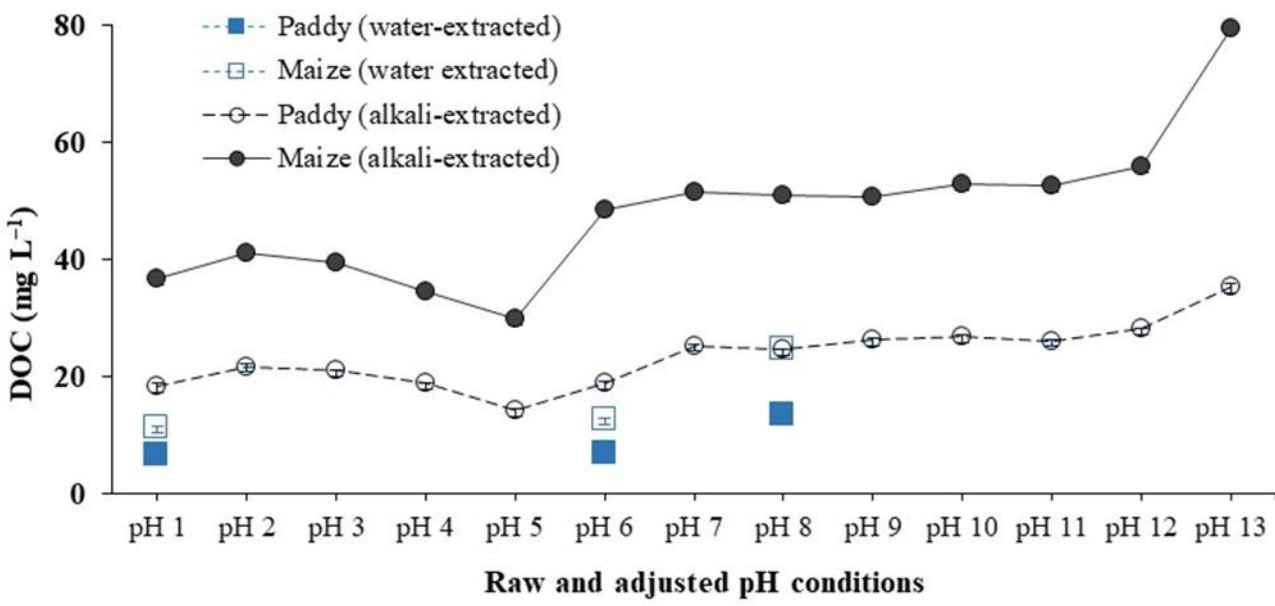

**Figure 3. Variation in DOC concetrations of the pH-adjusted HS$_{LS}$ and HS$_{CS}$ from paddy and maize soils.**

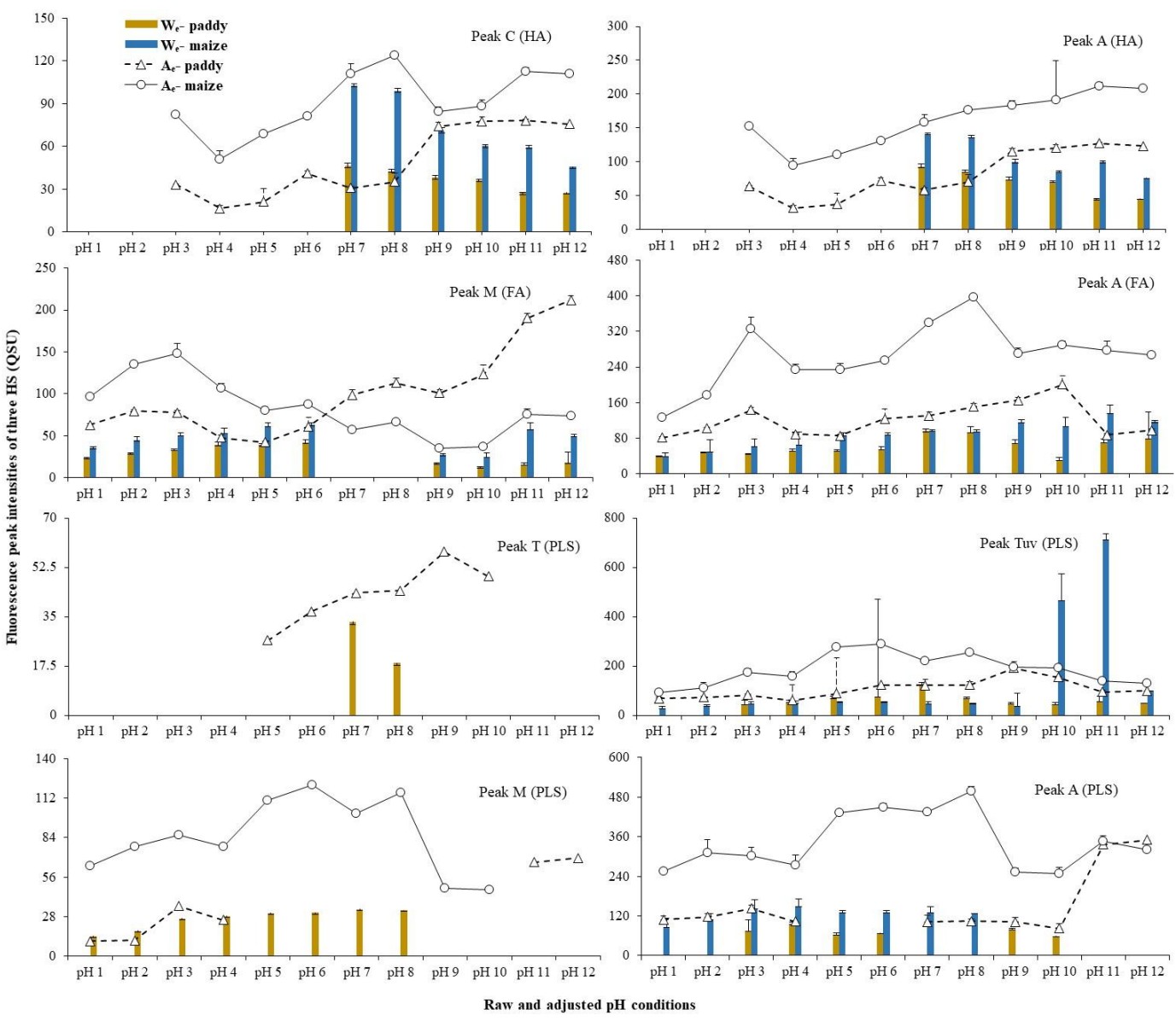

**Figure 4:** Fluorescence intensities of HA (peak C and peak A), FA (peak M and peak A) and PLS (peak T, peak T$_{UV}$, peak M and peak A) in pH-adjusted solutions of HS$_{LS}$ and HS$_{CS}$ from paddy and maize soils.

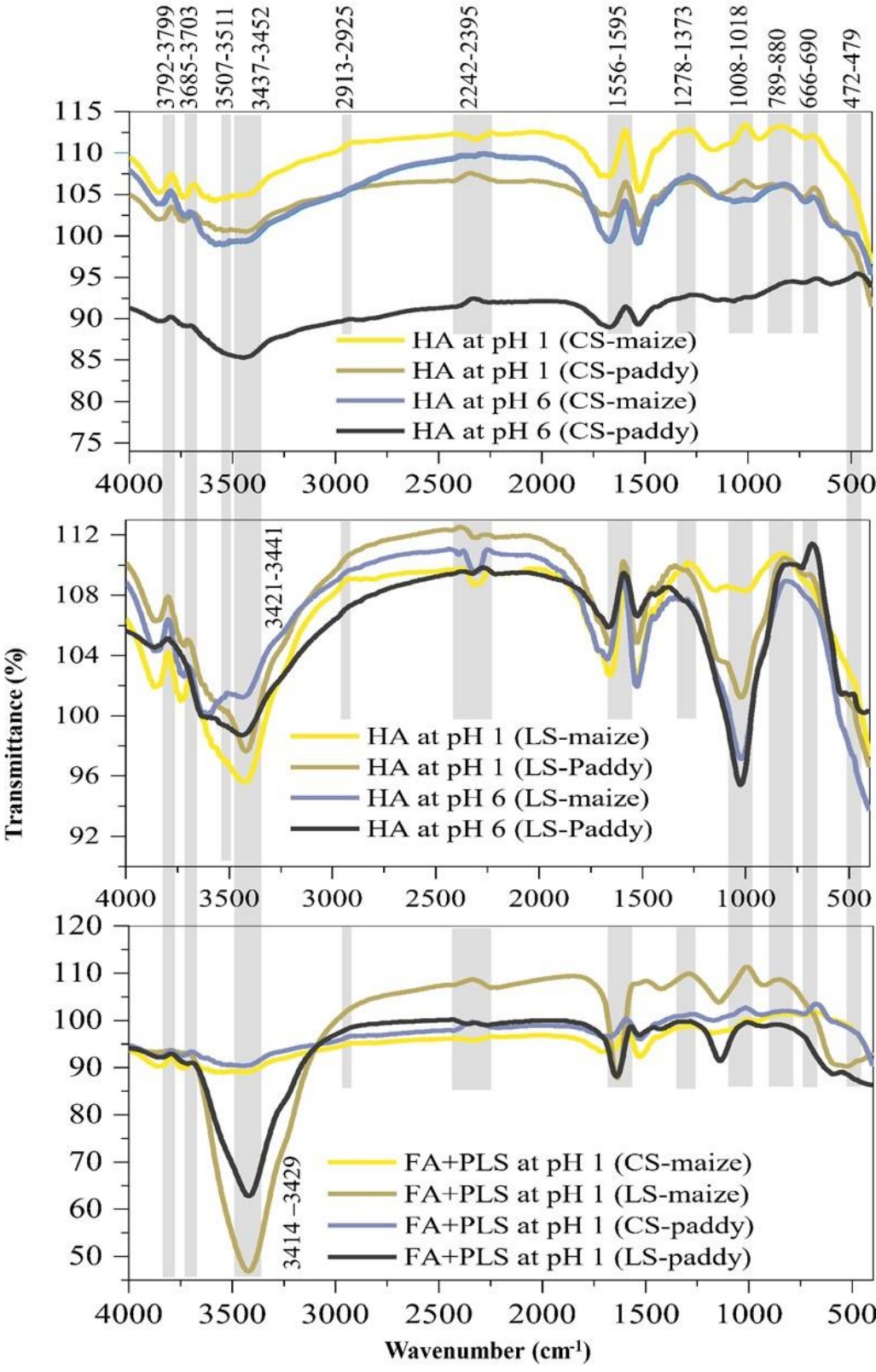

**Figure 5:** FTIR spectra of HA$_{LS-pH6}$, HA$_{LS-pH1}$, HA$_{CS-pH6}$, HA$_{CS-pH1}$, FA$_{LS}$+PLS$_{LS}$ at pH 1 and FA$_{CS}$+PLS$_{CS}$ at pH 1.

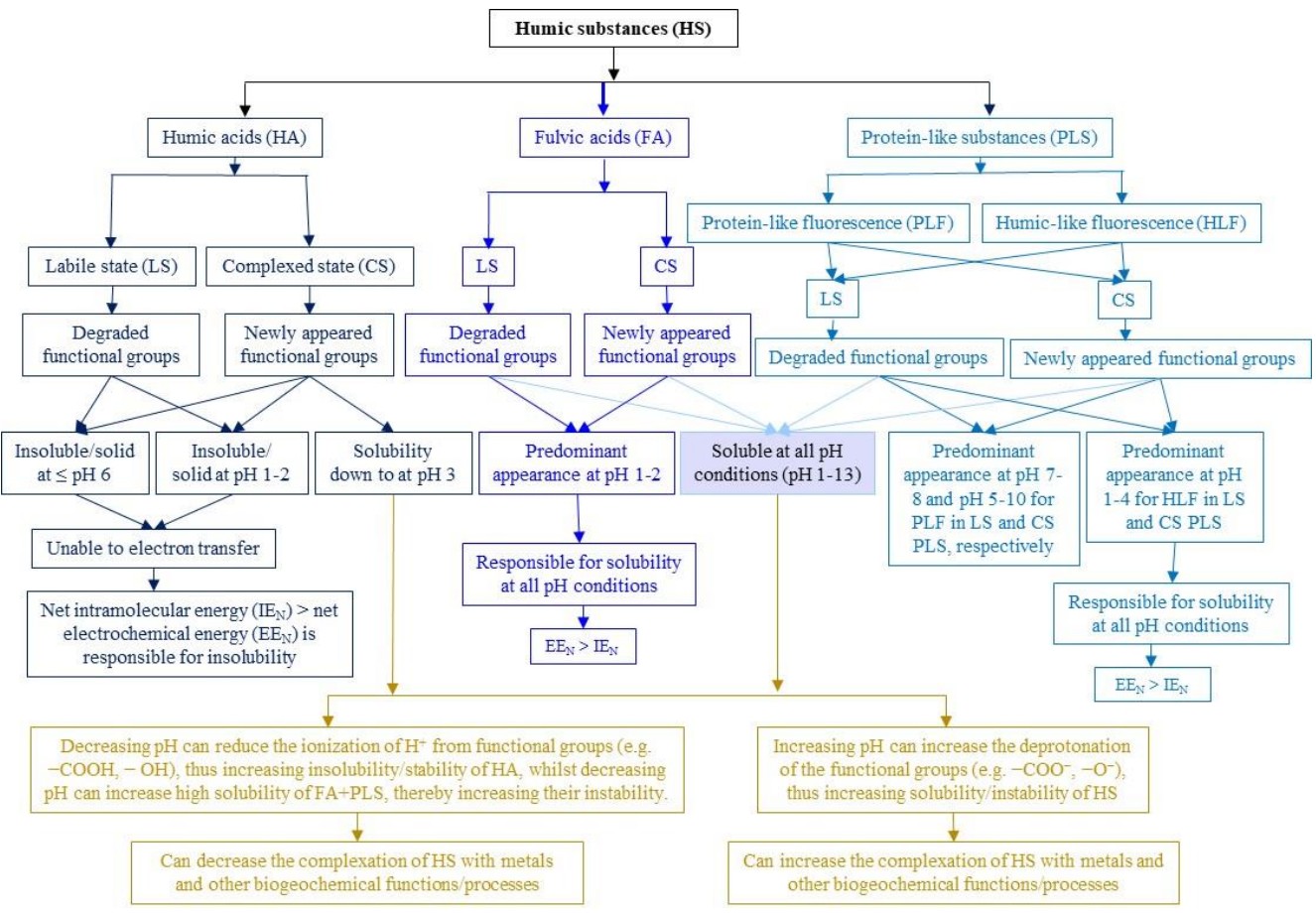

**Figure 6:** Conceptual model developed referring to $HS_{LS}$ and $HS_{CS}$, including HA, FA and PLS, based on the presence or absence of the corresponding fluorescence peaks in different pH conditions.

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
