# Peer review of "Solubility characteristics of soil humic substances as a 1"

_EGUsphere, 2023_

## Author Response (AR1)

**Referee comment#4**

The manuscript provides an insightful and methodologically robust exploration of the pH-dependent behaviors of humic substances (HS), including humic acids, fulvic acids, and protein-like substances in soils. By employing a combination of fluorescence excitation-emission matrix (EEM) spectroscopy and FTIR analysis, the study elucidates the solubility mechanisms of HS, revealing their dynamic interactions with environmental pH. These findings have important implications for understanding soil organic matter dynamics, sustainable soil management, and the global carbon cycle. Overall, the manuscript represents a valuable contribution to soil science and environmental studies.

Response: We are very grateful to the Reviewer for valuable and constructive comments on the manuscript. Thank you.

Itemized responses (R) to Reviewer's point by point comments are provided below.

**Major comments:**
Clarify whether adjustments in pH mimic natural soil conditions accurately. In general, soil pH range from 4-8, why you chose pH 1 to 12.
R: Based on the soil data summarized in Table S1, the pH of world soils vary from 2.80 to 9.39. As suggested, we have revised the text in **lines 138-143** as follows: 'Although the soil pH varies from 2.80 to 9.39 (Table S1), we fractionated the extracted solution in the pH range from 12 to 1 for several key reasons: Firstly, HS-bound as organo-minerals primarily liberate HS components and other constituents (e.g., various metals) in the liquid phase under alkaline extraction (0.1 M NaOH ≈ pH 13.0). Thus, it is crucial to understand how these HS components change their properties from pH 12 to 1. Secondly, it is essential to investigate how HS, in particular HA-DOC, due to its insoluble nature behave within the pH range from 12 to 1.'
Discuss how the findings may inform strategies for mitigating soil carbon loss.

R: We have revised the text in **lines 500-504** as follows: "7. The strategies for addressing pH-affected soil carbon loss primarily involve the significant loss of dissolved HS, particularly fulvic acids and protein-like fractions, from acidic soils due to water and rainwater runoff. Therefore, it is crucial to implement an efficient, rapid and sustainable drainage system that operates on a relatively short time scale, so that HS components may become less likely to dissolve in water and rainwater. Effective and timely drainage from the soil surface can help prevent carbon loss from acidic soils."

**Specific comments**
**Introduction:**
Consider elaborating on the novelty of this study compared to previous research to better establish its unique contributions.

R: We think that a short explanation regarding "the novelty of our study compared to previous research" is enough because reviewer's itself has been provided three more comments to add. Considering 'Introduction' in reasonably in good shape, we did not add in a more detailed

explanation in that regards. Besides, new additions from reviewer's three individual comments (1[st] major comment, and next two specific comments) are adequately added in the 'Introduction'. The previous studies explanation are mentioned below:

"Earlier studies (Hemingway et al., 2019; Lützow et al., 2006; Marschner et al., 2008; Sollins et al., 1996; Vogel et al., 2014) have not paid much attention to these issues when assessing the solubility and insolubility of SOM/HS. For example, pH effects were studied to assess the interaction mechanisms of Fe(II) ions with soil HA at pH values of 5 and 7 (Boguta et al., 2019), the binding of Cu and Pb to HA and FA at pH 4-8 (Christl et al., 2005), Cu(II) binding properties of soil FA at pH 7.0 (dos Santos et al., 2020), coagulation mechanisms of HA in metal ion solutions at pH 4.6-7.0 (Ai et al., 2020), coagulation behaviours of HA in $Na^+$ and $Mg^{2+}$ solutions at pH 3.6, 7.1, and 10.0 (Wang et al., 2013), and the disaggregation kinetics of peat HA at pH 3.65-5.56 (Avena and Wilkinson, 2002), but not directly in water and alkali-extracted soil HA, FA and PLS fractions. The acidic and alkaline pH conditions in the soil liquid phase alter the electronic configuration of the functional groups of HS components, which in turn affect their complexation capacity (Christl et al., 2005; Zhang et al., 2023; Avena and Wilkinson, 2002). The solubility and insolubility mechanisms of the HS components under different pH conditions remain unknown. In particular, two key fundamental questions regarding the effects of pH on HS are still unclear, that is, how the electrochemical behaviour of soil HS components changes in the pH range of 1–12, and how these changes affect the solubility/insolubility features of HS components and their mobilization/immobilization during rainwater runoff and groundwater infiltration in soil."

Please explain the practical importance of understanding the solubility of humic substances in relation to soil management or soil carbon dynamics.

**R:** As suggested by the Reviewer, we have added the following paragraph in ***lines 144-153*** of the revised Introduction: "Most importantly, the solubility of HS components and their subsequent mineralization are very relevant factor for the availability of soil nutrients and trace elements and the activity of soil microorganisms, while their stability in organo-minerals affects negatively these processes (Malik et al., 2018; Varghese et al., 2024; Lange et al., 1998; Gao et al., 2025; Soti et al., 2015; Yang et al., 2024; Gilbert et al., 2007; Zhang et al., 2023). These issues are concurrently associated with the corresponding biological fixation/sequestration of C, N and S by soil photosynthetic microorganisms (Green et al., 2019; Varghese et al., 2024; Heckman et al., 2001; Levicán et al., 2008; Ma et al., 2021; Kelly et al., 2021; Gao et al., 2025), and the subsequent release of extracellular polymeric substances and/or HS components in the neoformation of fresh organo-minerals in soil (Whalen et al., 2024; Yu et al., 2020; Paul, 2016; Kallenbach et al., 2016). Therefore, the solubility or insolubility of soil HS components is crucial for a better understanding of both soil management and soil carbon dynamics."

**References**

Gao, X., Zhang, J., Mostofa, K. M. G., Zheng, W., Liu, C. Q., Senesi, N., Senesi, G. S., Vione, D., Yuan, J, Liu, Y., Mohinuzzaman, M., Li, L., and Li, S. L.: Sulfur-mediated transformation, export and mineral complexation of organic and inorganic C, N, P and Si in dryland soils. Sci. Rep. under review (revising based on positive review comments), 2025.

Gilbert, B., Lu, G., and Kim, C. S.: Stable cluster formation in aqueous suspensions of iron oxyhydroxide nanoparticles, J. Colloid Interface Sci, 313, 152–159, 2007.

Green, J. K., Seneviratne, S. I., Berg, A. M. et al.: Large influence of soil moisture on long-term terrestrial carbon uptake, Nature, 565, 476–479, https://doi.org/10.1038/s41586-018-0848-x, 2019.

Heckman, D. S. et al.: Molecular Evidence for the Early Colonization of Land by Fungi and Plants. Science 293, 1129, 2001.

Kallenbach, C., Frey, S., and Grandy, A.: Direct evidence for microbial-derived soil organic matter formation and its ecophysiological controls, Nat Commun 7, 13630, https://doi.org/10.1038/ncomms13630, 2016.

Lange, O. L., Belnap, J., and Reichenberger, H.: Photosynthesis of the cyanobacterial soil-crust lichen Collema tenax

from arid lands in southern Utah, USA: Role of water content on light and temperature responses of $CO_2$ exchange. Funct. Ecol., 12, 195-202, 1998.

Levicán, G. et al.: Comparative genomic analysis of carbon and nitrogen assimilation mechanisms in three indigenous bioleaching bacteria: predictions and validations, BMC Genomics, 2008, 9, 581, 2008.

Ma, H. et al.: Rice Planting Increases Biological Nitrogen Fixation in Acidic Soil and the Influence of Light and Flood Layer Thickness, J. Soil Sci. Plant Nutr., 21, 341–348, 2021.

Malik, A. A., Puissant, J., Buckeridge, K. M., Goodall, T., Jehmlich, N., Chowdhury, S., et al.: Land use driven change in soil pH affects microbial carbon cycling processes, Nat. Commun. 9, 3591, https://doi.org/10.1038/s41467-018-05980-1, 2018.

Paul, E. A.: The nature and dynamics of soil organic matter: Plant inputs, microbial transformations, and organic matter stabilization, Soil Biol. Biochem., 98, 109-126, 2016.

Soti, P. G., Jayachandran K., Koptur, S., and Volin J.C.: Effect of soil pH on growth, nutrient uptake, and mycorrhizal colonization in exotic invasive Lygodium microphyllum, Plant Ecolog., 216, 989-998, 2015.

Varghese, E.M., et al.: Rice in acid sulphate soils: Role of microbial interactions in crop and soil health management. Appl. Soil Ecol., 196, 105309, 2024.

Whalen, E. D. et al.: Microbial trait multifunctionality drives soil organic matter formation potential. Nature Commun., 15, 10209, 2024.

Yang, X., Gao, X., Mostofa, K. M. G., Zheng, W., Senesi, N., Senesi, G. S., Vione, D., Yuan, J., Li, S. L., Li, L., Liu, C. Q.: Mineral states and sequestration processes involving soil biogenic components in various soils and desert sands of Inner Mongolia, Sci. Rep. 14, 28530, https://doi.org/10.1038/s41598-024-80004-1, 2024.

**Methods:** The extraction protocols are clearly described, but providing a brief rationale for selecting the specific soil types (paddy and maize soils).

**R:** As suggested by the Reviewer, we have added the following text in *lines 173-178*: "Importantly, the rationale for selecting paddy and maize soils is based on their distinct characteristics, i.e., paddy soils are submerged for extended periods, while maize soils are relatively less influenced by the presence of water. Therefore, HS components of these two soil types are expected to be altered very differently in their organo-mineral lability and stability in the pH range from 1 to 12. This study is expected to provide useful information on soil carbon dynamics and contribute to minimize soil carbon loss during agricultural practices."

Discussion: Expand on why understanding the molecular-level solubility is critical for agricultural practices.

**R:** As suggested by the Reviewer, we have added the following text in **lines 482-499**: "6. The knowledge of the molecular-level solubility of the three HS components is essential for a better understanding and management of agricultural practices, as affected by their individual solubility, tendency to precipitate, and variable capability to form organo-minerals, which can occur more or less rapidly/slowly (Underwood et al., 2024; Zhang et al., 2023). For instance, the HA fractions of acidic soils may either partially precipitate or remain in suspension due to an increase in $IF_N$ from increasing intramolecular interactions among various functional groups via hydrogen bonding, influenced by acidic conditions as discussed earlier. As a result, precipitated HA fractions would enhance C stability, while suspended HA fractions are highly prone to leaching by rainwater runoff. Furthermore, the high solubility of FA and PLS under acidic conditions would result from prolonged water saturation occurring in paddy fields, which will lead to soil C loss by their transport due to rainwater runoff. Simultaneously, these conditions of HS are expected to contribute to increase the salinity levels in such type of soils (Varghese et al., 2024; Ma et al., 2021). Therefore, high-water-demand crops such as rice may not be suitable for

maintaining C stability in acidic soils. In contrast, low-water-demanding crops like maize and wheat would be more effective in minimizing C loss from acidic soils. On the other hand, alkaline soils can support the cultivation of a wide variety of crops while minimizing C loss, along with the presence of relatively high levels of HS-bound to organo-minerals. In particular, the pH levels of the paddy and maize soils object of this study are slightly alkaline (8.13 and 7.92, respectively), making them reasonably suitable for diverse types of crops. Furthermore, both soils exhibit relatively high levels of HS-bound organo-minerals, with significant increases in $DOC_{CS}$ stability (2.6 and 3.2 times higher, respectively) compared to $DOC_{LS}$ lability."

**New references**
Underwood, T. R. et al.: Mineral-associated organic matter is heterogeneous and structured by hydrophobic, charged, and polar interactions. PNAS 121, e2413216121 https://doi.org/10.1073/pnas.2413216121, 2024.

Conclusion: Conclusion is repetitive. Please consider addressing any future research directions.

**R:** As suggested, we have revised the repetition in lines 516-525 as follows (blue color new addition):
"In particular, an alkaline or elevated pH level would result in anionic forms ($-O^-$ and $-COO^-$) of phenolic OH and carboxyl groups of HA, FA and PLS, which ultimately contributes to the insolubilisation and stability of HS through the formation of organo-mineral complexes in soils. In contrast, at acidic pH, the electron and proton transfer processes would be facilitated by the availability of uncomplexed metal ions, with subsequent insolubility of $HA_{LS+CS-pH6}$ which would remain insoluble in soils during rainwater events or water runoff at pH 6, whereas $HA_{LS+CS-pH1}$ would remain soluble and thus mobile and would be transported in ambient surface waters via rainwater, leaching, and groundwater infiltration (Ronchi et al., 2013; Stolpe et al., 2013; Mostofa et al., 2019). The highly soluble FA and PLS at acidic conditions would facilitate to the easy transport to ambient surface waters via rainwater and groundwater discharge (Ronchi et al., 2013; Stolpe et al., 2013; Mostofa et al., 2019)."

We have added the future research direction the following text in lines 531-533: "Future research directions should focus on investigating acidic soils, which are beyond the scope of this study. These soils are expected to be significantly affected by the individual conditions of HA, FA, and PLS in acidic environments."

**Line 60:** Define "FTIR" in the first instance it appears for readers who may not be familiar with the term.

**R:** As suggested, revised in line 68, 240 and 391

**Line 110-115**: Expand on the significance of fluorescence excitation-emission matrix (EEM) and its advantages over other methods.

**R:** As suggested, we have added the following text of the revised Introduction in *lines 99-105*: 'Three-dimensional (3D) fluorescence excitation-emission matrix (EEM) spectroscopy (3D EEMS) is a precise, rapid and relatively simple technique for measuring filtered environmental surface waters and samples extracted from soils and sediments (Senesi, 1990b; Coble, 1990, 1996; Stedmon et al., 2003; Mostofa et al., 2013; Mohinuzzman et al., 2020). In particular, this technique allows the characterization of fluorescent components, including soil HS, autochthonous humic-like substances, PLS, detergent-like substances, and others, without the need for further pretreatment of the samples (Senesi, 1990b; Coble 1990, 1996; Stedmon et al., 2003; Mostofa et al., 2013).

**New references**
Coble, P. G., Green, S. A., Blough, N. V., and Gagosian, R. B.: Characterization of dissolved organic matter in the Black Sea by fluorescence spectroscopy, Nature, 348, 432–435, 1990.

Coble, P. G.: Characterization of marine and terrestrial DOM in sea water using excitation-emission matrix spectroscopy, Mar. Chem., 52, 325–346, 1996.

Mostofa, K. M. G., Yoshioka, T., Mottaleb, M. A., Vione, D.: Photobiogeochemistry of Organic Matter: Principles and Practices in Water Environments. Springer, Berlin, Germany, 2013.

Senesi, N.: Molecular and quantitative aspects of the chemistry of fulvic acid and its interactions with metal ions and organic chemicals. Part II. The fluorescence spectroscopy approach, Anal. Chim. Acta, 232, 77–106, 1990a.

Stedmon, C. A., Markager, S. and Bro, R. (2003) Tracing dissolved organic matter in aquatic environments using a new approach to fluorescence spectroscopy. *Mar. Chem.* **82**, 239–254

**Line 130-135**: Include a brief justification for selecting maize and paddy soils as the study sites.
More references for major FTIR absorption bands and assignments.
Use consistent formatting for citations throughout the text.

**R:** As suggested, for the first issue (and also referee's same previous comment), we have already added this in lines 173-178.

For the second issue, we have added more relevant references regarding FTIR band peaks in lines 403-416

For the third issue, we have carefully checked the consistent formatting for citations throughout the text.

**New references**:
Demyan, M. S. et al.: Use of specific peaks obtained by diffuse reflectance Fourier transform mid-infrared spectroscopy to study the composition of organic matter in a Haplic Chernozem, Eur. J. Soil Sci., 63, 189–199, 2012.
Kunlanit, B., Vityakon, P., Puttaso, A., Cadisch, G., and Rasche, F.: Mechanisms controlling soil organic carbon composition pertaining to microbial decomposition of biochemically contrasting organic residues: Evidence from midDRIFTS peak area analysis, Soil Bio. Biochem. 76, 100–108, 2014.

Shammi, M., Pan, X., Mostofa, K. M. G., Zhang, D., Liu, C. Q.: Photo-flocculation of algal biofilm extracellular polymeric substances and its transformation into transparent exopolymer particles. Chemical and spectroscopic evidences. Sci. Rep. 7, 9074, 2017.

Singh, R. P. et al. Isolation and characterization of exopolysaccharides from seaweed associated bacteria Bacillus licheniformis. Carbohyd. Polym. 84, 1019–1026 (2011).

**Citation**: https://doi.org/10.5194/egusphere-2023-2994-RC2